# The molecular dissection of TRIM25's RNA-binding mechanism provides key insights into its antiviral activity

Lucía Álvarez [1,16], Kevin Haubrich [1,16], Louisa Iselin[2,3,4], Laurent Gillioz [5], Vincenzo Ruscica [4], Karine Lapouge [6], Sandra Augsten[1], Ina Huppertz[7], Nila Roy Choudhury[8,9], Bernd Simon[1,10], Pawel Masiewicz[1], Mathilde Lethier[11], Stephen Cusack [11], Katrin Rittinger [12], Frank Gabel [13], Alexander Leitner [14], Gracjan Michlewski [8,9], Matthias W. Hentze [7], Frédéric H. T. Allain [5], Alfredo Castello [4] ✉ & Janosch Hennig [1,15] ✉

TRIM25 is an RNA-binding ubiquitin E3 ligase with central but poorly understood roles in the innate immune response to RNA viruses. The link between TRIM25's RNA binding and its role in innate immunity has not been established. Thus, we utilized a multitude of biophysical techniques to identify key RNA-binding residues of TRIM25 and developed an RNA-binding deficient mutant (TRIM25-m9). Using iCLIP2 in virus-infected and uninfected cells, we identified TRIM25's RNA sequence and structure specificity, that it binds specifically to viral RNA, and that the interaction with RNA is critical for its antiviral activity.

The innate immune system is the body's first line of defence against viral infection. When viruses are detected by the host cell, signalling pathways are activated to induce type I interferon response and the production of inflammatory cytokines[1]. Post-translational modifications regulate these signalling pathways, allowing a dynamic and specific host response to infection[2]. Ubiquitination plays a particularly important role and multiple E3 ubiquitin ligases have been identified to be involved in the regulation of immunity[3–6]. The Tripartite Motif (TRIM) family of E3 ligases is characterized by their domain architecture, consisting of an N-terminal RING domain, which is responsible for the E3-ligase activity, followed by one or two B-box domains, a coiled-coil (CC) dimerisation domain and a C-terminal region containing non-catalytic domains that recognise E3 ligase substrates and other targets, which are variable across TRIM proteins with TRIM25 featuring a PRY/SPRY domain (Fig. 1a) [7–9]. TRIM25 has been identified as a key player in triggering the innate immune response to RNA viruses. It has been shown that TRIM25 plays a role in the RIG-I signalling pathway, where it is proposed to ubiquitinate the RIG-I

[1]Molecular Systems Biology Unit, European Molecular Biology Laboratory (EMBL) Heidelberg, 69117 Heidelberg, Germany. [2]Nuffield Department of Medicine, Peter Medawar Building for Pathogen Research, University of Oxford, Oxford OX1 3SY, UK. [3]Department of Biochemistry, University of Oxford, South Parks Road, OX1 3QU Oxford, UK. [4]MRC-University of Glasgow Centre for Virus Research, 464 Bearsden Road, Glasgow G61 1QH Scotland, UK. [5]Institute of Biochemistry, Department of Biology, ETH Zürich, Zürich, Switzerland. [6]Protein expression and purification facility, European Molecular Biology Laboratory (EMBL) Heidelberg, 69117 Heidelberg, Germany. [7]Director's Research, European Molecular Biology Laboratory (EMBL) Heidelberg, 69117 Heidelberg, Germany. [8]Dioscuri Centre for RNA-Protein Interactions in Human Health and Disease, International Institute of Molecular and Cell Biology in Warsaw, Warsaw, Poland. [9]Infection Medicine, University of Edinburgh, The Chancellor's Building, Edinburgh, UK. [10]Department of Molecular Biology and Biophysics, University of Connecticut Health Center, Farmington, CT, USA. [11]European Molecular Biology Laboratory, 71 Avenue des Martyrs, CS 90181, 38042 Grenoble Cedex 9, Grenoble Cedex, France. [12]Molecular Structure of Cell Signalling Laboratory, The Francis Crick Institute, 1 Midland Road, London NW1 1AT, UK. [13]Université Grenoble Alpes, Institut de Biologie Structurale, Grenoble, France; Commissariat à l'Energie Atomique et aux Energies Alternatives, Direction de la Recherche Fondamentale, Institut de Biologie Structurale, Grenoble, France; Centre National de la Recherche Scientifique, Institut de Biologie Structurale, Grenoble, France. [14]Institute of Molecular Systems Biology, Department of Biology, ETH Zürich, 8093, Zürich, Switzerland. [15]Chair of Biochemistry IV, Biophysical Chemistry, University of Bayreuth, 95447 Bayreuth, Germany. [16]These authors contributed equally: Lucía Álvarez, Kevin Haubrich. ✉e-mail: alfredo.castello@glasgow.ac.uk; janosch.hennig@embl.de

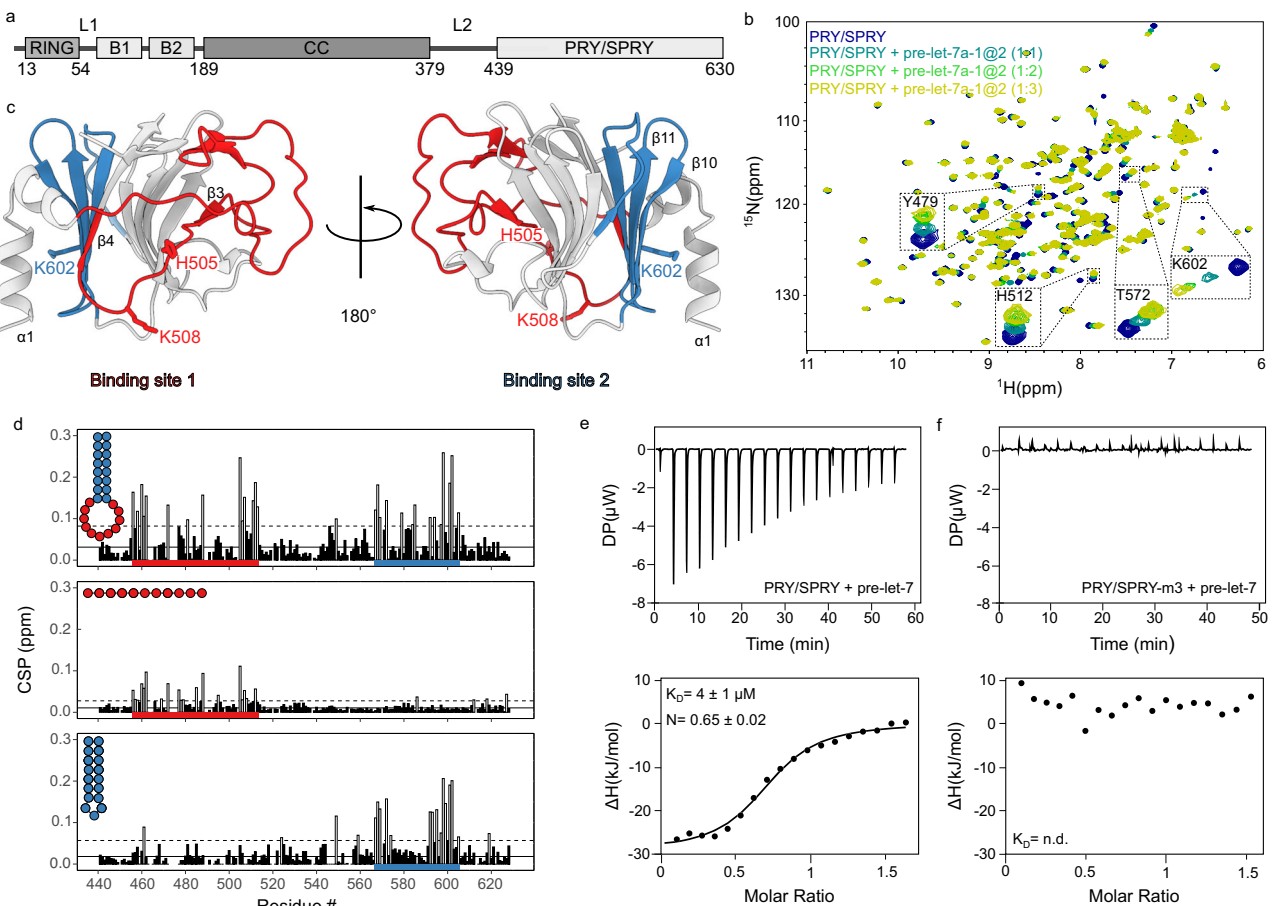

**Fig. 1 | NMR and ITC analysis of RNA binding by TRIM25 PRY/SPRY domain.**
**a** Domain arrangement of TRIM25. **b** $^1$H-$^{15}$N-HSQC spectra of 100 μM $^{15}$N-labelled TRIM25 PRY/SPRY domain free (dark blue) and in presence of different ratios of pre-let-7 RNA (1:1 light blue, 1:2 green and 1:3 yellow). **c** Significant CSPs from the titration of TRIM25 PRY/SPRY with pre-let-7 plotted on the structure (PDB: 6FLM) indicate two binding sites (binding site 1 coloured red, binding site 2 coloured blue). The structure shown was generated using UCSF ChimeraX[80]. **d** Histograms of chemical shift perturbations (CSPs) for TRIM25·PRY/SPRY upon binding to full-length pre-let-7 (top), pre-let-7-loop (middle) and pre-let-7-stem (bottom). The continuous line shows the average CSP and the dashed line indicates the average plus one standard deviation. Residues for which the CSPs are more than one standard deviation above the average are shown in open bars. Significantly affected regions are indicated in the protein sequence by red and blue horizontal bars. The RNA is coloured according to where it binds in the PRY/SPRY domain (binding site 1 in red, binding site 2 in blue). Source data are provided as a Source Data file. **e** Representative ITC binding isotherm for TRIM25 PRY/SPRY:pre-let-7 complex ($n = 4$). The value shown in the figure is the average of all replicates and its standard deviation. **f** Representative ITC binding isotherm for TRIM25 PRY/SPRY triple mutant (m3): pre-let-7 complex ($n = 3$). All experimental setups and results of ITC measurements including replicates can be found in Supplementary Table 1.

caspase activation and recruitment domains (CARDs), which are exposed upon recognition of viral 5′-triphosphate-blunt-end double-stranded RNA[10,11]. However, there are other E3 ubiquitin ligases that can polyubiquitinate and activate RIG-I, such as RING finger protein leading to RIG-I activation (Riplet)[12], mex-3 RNA binding family member C (MEX3C)[13] and TRIM4[14]. In addition, there are studies suggesting that only RIPLET and not TRIM25 is sufficient to ubiquitinate and activate RIG-I[15–17]. Despite these inconsistencies, TRIM25's involvement in antiviral activity is undisputed. Apart from the RIG-I pathway TRIM25 is required for activating the antiviral function of zinc-finger antiviral protein (ZAP)[18,19], which inhibits the replication of a wide-range of viruses by binding to viral mRNA, repressing its translation and promoting its degradation[19–23].

TRIM25 was identified as an RNA-binding protein (RBP) by several RNA interactome capture studies, which reported peptides that crosslink to RNA[24–26]. Several studies could identify different regions of the protein, which harboured RNA-binding residues (located in the CC, PRY/SPRY domains, and the linker in between[25,27–29]). TRIM25 has also been shown to interact with a wide variety of RNAs, including 3′-UTRs and exons of mRNAs[25,28], lincRNAs[28], miRNAs[28], viral RNAs and their corresponding ribonucleoproteins[30,31]. Although no specific consensus

motif for RNA binding was identified, G- and C-rich sequences were found to be overrepresented in the CLIP-seq data[28]. In vitro assays showed no clear preference for single- or double-stranded RNA[29]. The fact that different domains and regions of TRIM25 seem to be involved in RNA binding makes it difficult to identify specific sequences or conformational RNA motifs that TRIM25 prefers, and points towards a structurally intricate RNA-binding mechanism. Moreover, it was shown that TRIM25 required RNA binding for its E3-ligase function[28,29] and overall antiviral activity[29]. However, the mutants employed in those studies still retained RNA-binding activity or were partially unfolded, as we demonstrate below. Thus, it remains poorly understood how TRIM25 binds RNA and how this affects its antiviral activity.

Here we combine biochemical, biophysical and cell biology approaches to characterise the TRIM25-RNA interaction and its role during antiviral activity. We have identified key residues in the PRY/SPRY and CC domains responsible for RNA binding. Furthermore, we developed a mutant, which retains the protein structure including an intact RING domain, responsible for E3-ligase activity, but completely lacks RNA-binding ability. Using iCLIP2[32] with size matched inputs in virus-infected and uninfected cells, we could identify the host and viral RNA motifs that TRIM25 binds to and determined TRIM25's

RNA structure- and sequence specificity. In addition, we could confirm that RNA binding of TRIM25 is required for its antiviral activity as well as for its subcellular redistribution to the viral replication organelles (ROs) where it co-localises with viral RNA.

## Results

### TRIM25 PRY/SPRY domain binds to single- and double-stranded RNA

Previous work showed that the PRY/SPRY domain is responsible for TRIM25 RNA binding[25,28,29,33]. Choudhury et al. identified a region in the PRY/SPRY domain encompassing residues 470–508 that bind to RNA by screening a series of deletion constructs[28]. However, deletion of this region unfolds the PRY/SPRY domain and therefore it remained unclear whether impairment of RNA binding is due to removal of RNA-binding residues or unfolding of the domain (Supplementary Fig. 1a). To refine the RNA-binding interface of this domain at residue-level resolution, we performed NMR titrations with the reported RNA target of TRIM25, pre-let-7a-1@2[34] (hereafter referred to as pre-let-7, Supplementary Fig. 1b). NMR resonances of residues 439–630 of the PRY/SPRY were assigned based on a previously published assignment[35]. Upon addition of pre-let-7, chemical shift perturbations (CSPs) are observed, which confirms binding (Fig. 1b). Specifically, we noticed strong CSPs clustered around two regions of the PRY/SPRY domain. The first of the two regions (region 1, aa. 456–513) is located at the C-terminus of the PRY motif, with residues H505 and K508 in the flexible loop connecting β-strands 3 and 4 exhibiting the strongest CSPs (Fig. 1c, d). This region is located close to the previously reported interaction site between the PRY/SPRY and CC domains[35]. The second region (region 2, amino acids 567–605) consists of β-strands 10 and 11, close to the N-terminal helix α1, with residue K602 having the strongest CSPs (Fig. 1c, d). At this stage, we could not assess, whether the CSPs in this region were due to a direct RNA interaction or due to an allosteric effect.

The predicted structure of pre-let-7 consists of a stem-loop (Supplementary Fig. 1b). The formation of double-stranded regions indicative of a stem structure was confirmed by the presence of imino peaks detected in 1D-$^1$H and 2D-$^1$H/$^1$H-NOESY experiments (Supplementary Fig. 1c). Imino signals are only observable upon base pairing as the exchange of imino protons with solvent is too fast when unpaired. To obtain further insights into the RNA-binding mechanism of the PRY/SPRY, we designed shorter RNA constructs consisting of only the loop (pre-let-7-loop, single-stranded RNA) or stem of pre-let-7 (pre-let-7-stem) (Supplementary Fig. 1b). To ensure that the short stem was double-stranded, we fused the strands through a three-base long linker and confirmed its stem formation in solution by $^1$H/$^1$H-NOESY NMR (Supplementary Fig. 1c). Pre-let-7-loop induced CSPs only for residues located in binding site 1, whereas titration of pre-let-7 stem affected only binding site 2 (Fig. 1d). To confirm that the binding sites are generally involved in RNA binding and are not specific for pre-let-7 RNA, the NMR titration experiments were repeated with other known RNA binders of TRIM25 (such as Lnczc3h7a[36], DENV[30] and an arbitrary 28-mer[29], see Supplementary Fig. 1b) with similar outcome (Supplementary Fig. 1d). Interestingly, these RNAs also consist of stem-loop structures confirmed by 1D-$^1$H and $^1$H/$^1$H-NOESY experiments (Supplementary Fig. 1c). With Lnczc3h7a, nuclear Overhauser effect (NOE) cross-peaks are not visible due to the low sample concentration. Nevertheless, the presence of imino peaks in the diagonal and 1D experiments confirms base pairing. These results suggest that the PRY/SPRY domain interacts with RNA by two distinct binding sites with different structural specificities. The first binding site interacts with single-stranded RNA and the second with double-stranded RNA. The structural similarity between the RNAs reported to bind to TRIM25 suggests a preference for stem-loop structures[29,30,36].

The PRY/SPRY domain interacts with different stem-loop RNAs with a dissociation constant ($K_D$) in the low micromolar range

(0.4−4 μM), as assessed by isothermal titration calorimetry (ITC, see Fig. 1e, Supplementary Fig. 1e and Supplementary Table 1). This is in the same range as canonical RNA-binding domains such as RNA recognition motifs (RRMs)[37]. Our ITC data indicate that the PRY/SPRY domain preferentially binds to loops rich in A and G, as we could detect binding to pre-let-7-loop but not to the Lnczc3h7-loop, which has three U and is shorter (Supplementary Table 1 and Supplementary Fig. 1b). The $N$ values obtained from our ITC experiments for PRY/SPRY binding to its target RNAs was consistently around 0.3, which implies a stoichiometry of approximately 1 RNA per 3 PRY/SPRY domains. Despite careful control of reactant concentrations in repeated experiments, the observed $N$ values persisted, ruling out concentration-derived artifacts and supporting a complex binding mechanism.

To further characterise the RNA-binding site of the PRY/SPRY, we mutated the three residues exhibiting the largest CSPs in NMR titrations. We created a triple mutant, termed PRY/SPRY-m3, in which we mutated residues H505 and K508 from binding site 1 and residue K602 from binding site 2 to glutamic acids. Importantly, the $^1$H,$^{15}$N-HSQC spectrum of the PRY/SPRY-m3 confirmed that the mutated PRY/SPRY domain is properly folded (Supplementary Fig. 1f). ITC showed that the PRY/SPRY-m3 lost its ability to interact with RNA, further confirming the importance of these residues for RNA binding (Fig. 1f and Supplementary Table 1). The occurrence of these RNA-binding residues seems to be a unique feature for TRIM25 amongst human TRIM family members from group IV[9] (i.e.,TRIM proteins carrying a PRY/SPRY domain) revealed by multiple sequence alignments. However, the RNA-binding residues are highly conserved in TRIM25 across different species (Supplementary Fig. 1g, h and Supplementary Table 2).

### CC domain binds RNA

Having characterised RNA-binding activity of the PRY/SPRY domain, we followed up on previous work that has reported the CC domain of TRIM25 as RNA binder[25,28,38]. The CC is not suitable for NMR studies due to its size and extended conformation, which causes slow molecular tumbling and NMR resonance line broadening beyond detection (Supplementary Fig. 2a). We therefore used ITC to test the putative RNA-binding capacity of the CC domain. We found that the CC interacts with pre-let-7 with low micromolar affinity (3.2 ± 0.7 μM) (Fig. 2a), in the same range as the PRY/SPRY domain and classical RNA-binding domains (e.g., RRMs). We performed ITC measurements with the other RNAs used in this study (Supplementary Figs. 1b and 2b). In contrast to the PRY/SPRY domain, the CC domain binds pre-let-7 or lnczc3h7a with equal affinities, but binds 10-fold weaker to the isolated loop sequence of both RNAs (Supplementary Table 1, and Supplementary Fig. 2b). Thus, we find that the CC domain is more promiscuous than the PRY/SPRY domain in its sequence preference.

Since we were unable to map the RNA-binding interface on the CC by NMR, we relied on indirect information to design CC RNA-binding deficient mutants. Based on the CC-PRY/SPRY structure[35], we deduced the RNA-binding residues from amino acids that are surface-exposed, close to the CC:PRY/SPRY interface and are commonly found to interact with RNA, such as lysines, arginines and aromatic residues. We found two candidates for mutational analysis (K283 and K285, Fig. 2b) on the basic surface close to the PRY/SPRY binding site 1. Mutation of these two residues into alanines (CC-m2) had a minor effect on affinity towards pre-let-7. However, this reduced the binding enthalpy 10-fold (Fig. 2c and Supplementary Table 1), while retaining the ability to dimerize and to fold into a coiled coil (Supplementary Fig. 2c, d). These findings suggest that while both amino acids are involved in energetically favourable interactions with RNA, other residues must contribute to it.

Using CLIR-MS[39] (Fig. 2d) with an equimolar mixture of unlabelled and uniformly $^{13}$C,$^{15}$N-labelled 85-nucleotide-long RNA incubated in HeLa nuclear extract, we detected UV-crosslinks between nucleotides

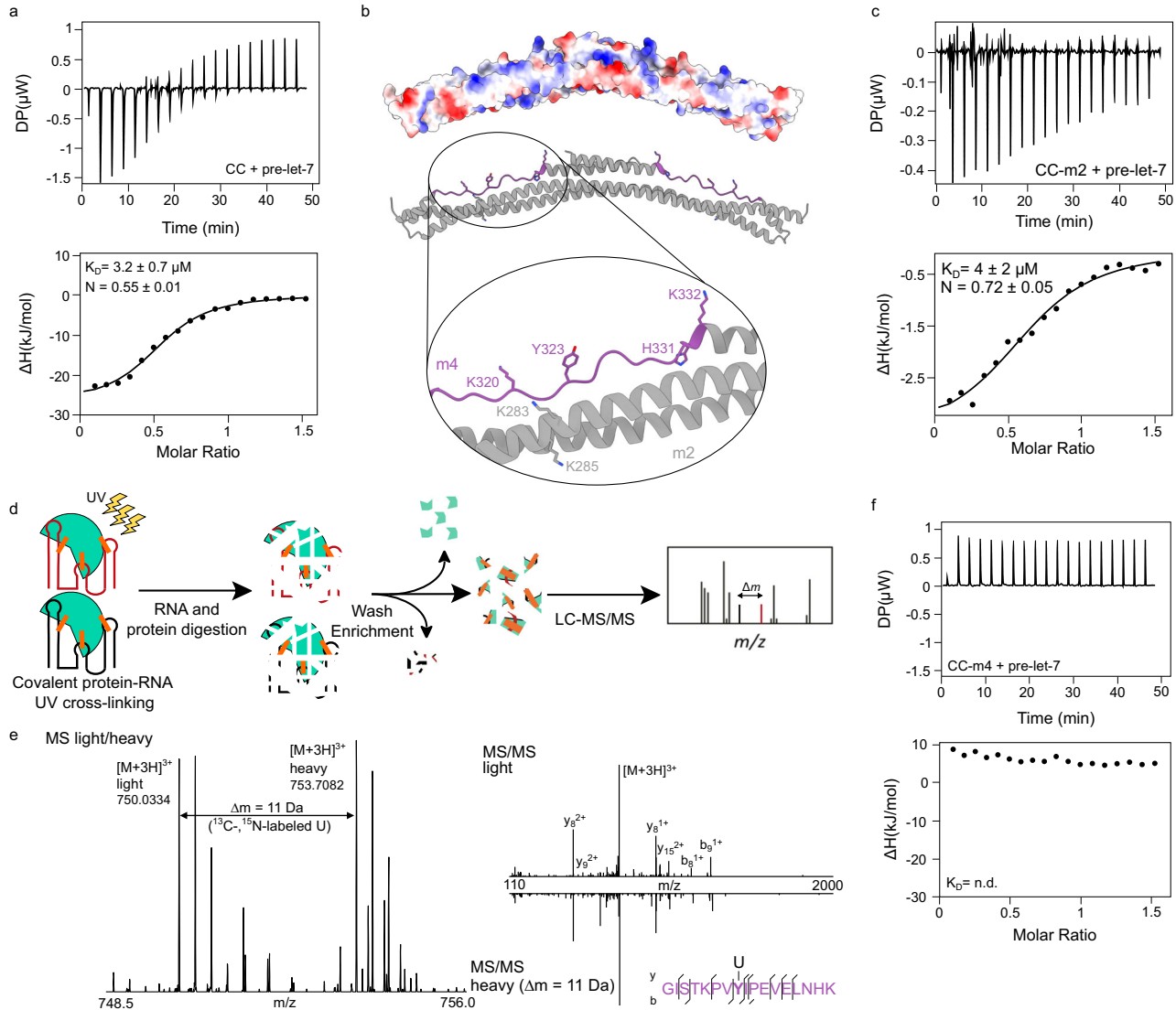

**Fig. 2 | ITC and CLIR-MS/MS analysis of RNA binding by TRIM25 CC domain.**
**a** Representative ITC binding isotherm for TRIM25 CC:pre-let-7 complex ($n = 3$).
The value shown in the figure is the average of all replicates and its standard
deviation. **b** The surface potential representation of the CC dimer (PDB:4LTB)
indicates a positively charged surface. The enlargement of this interface shows
potential RNA binding residues. Mutation of residues K283 and K285, shown in
grey, leads to a TRIM25-CC mutant we term m2. In purple, the peptide detected by
CLIR-MS/MS (see below) is highlighted including the four potential RNA binding
residues. Mutation of these four residues leads to a TRIM25-CC mutant we term m4.
The structure shown was generated using UCSF ChimeraX[80]. **c** Representative ITC
binding isotherm for TRIM25 CC-m2 with pre-let-7 complex ($n = 3$). **d** Schematic
representation of the CLIR-MS/MS method. The RNA binding protein (grey) is
crosslinked to an equimolar mixture of unlabelled (black) and uniformly
$^{13}$C/$^{15}$N-labelled RNA (red). **e** MS spectrum of the TRIM25-derived peptide GISTKP-
VYIPEVELNHK crosslinked to a single U nucleotide (see methods). **f** Representative
ITC binding isotherm for TRIM25 CC-m4:pre-let-7 complex ($n = 3$). All experimental
setups and ITC measurements including replicates are listed in Supplementary
Table 1.

and amino acids of the peptide GISTKPVYIPEVELNHK of TRIM25
(Fig. 2e). The MS/MS spectrum allowed the identification of the
crosslinked residue, which was either tyrosine 323 (Y323) or the neigh-
bouring isoleucine 324 (I324). The peptide crosslinks to one uridine
(mass shift 324) and could be identified by "light" searches (based on
the expected mass of the RNA-peptide adduct) and by "light-heavy"
searches (considering the characteristic peak doublet resulting from the
isotope labelling), ensuring the specificity for the differentially isotope
labelled RNA (Fig. 2e). According to a crystal structure (PDB: 6FLN[35]) this
region is devoid of secondary structure (Fig. 2b). Comparative analysis
of TRIM25 sequences from different species revealed that the specific
region responsible for RNA binding, as identified by CLIR-MS, is well
conserved (see Supplementary Table 2 and Supplementary Fig. 2e).
However, this region is not conserved in other group IV TRIM protein CC
domains[9] (Supplementary Fig. 2f). It is noteworthy that this region

overlaps with the surface that interacts with the PRY/SPRY domain
identified by Koliopoulos et al. [35] (Fig. 2b). To confirm that this region
interacts with RNA, we mutated these four residues (K320, Y323, H331
and K332) to glutamic acids (CC-m4) and performed ITC experiments.
These show that the CC-m4 has lost its RNA-binding activity (Fig. 2f),
while still retaining its ability to fold into a coiled coil and dimerize
(Supplementary Fig. 2c, d). We have thus identified the residues of the
CC domain that are critical for RNA binding.

**PRY/SPRY and CC domains bind to RNA cooperatively**
As both domains, the CC and PRY/SPRY interact with RNA, we wanted
to assess, whether they do so cooperatively. To test this, we performed
ITC measurements of the CC-PRY/SPRY domain. The affinity of the
CC-PRY/SPRY construct to pre-let-7 RNA is more than 30–50-fold
higher than that of the isolated domains (Fig. 3a), which is a hallmark of

chelating cooperativity, where multiple individually weaker binary interactions cooperate to form a stable multimeric complex[40,41]. This cooperative effect can also be observed for binding to Lnzc3h7a RNA (Supplementary Table 1 and Supplementary Fig. 3a). It should be highlighted that the ITC binding curves for Lnzc3h7a and DENV-SL with CC-PRY/SPRY have a biphasic behaviour, in contrast to the monophasic curve observed for pre-let-7. This could be due to various factors such as conformational change of the RNA[42].

We have previously shown that the interaction between the CC and PRY/SPRY domains in solution is rather transient, despite being present in the crystal structure of the CC-PRY/SPRY tandem domain construct (PDB: 6FLN [35]). Considering that the CC-PRY/SPRY interaction plays an important role in TRIM's E3 ligase activity and that the identified RNA binding residues are close to the CC:PRY/SPRY interaction site[35], we investigated whether RNA binding modifies the properties of this interaction in solution. To achieve this, we monitored the changes in $^1$H,$^{15}$N-HSQC NMR spectra of CC-PRY/SPRY upon addition of pre-let-7. In the absence of RNA, the spectrum is dominated by sharp, high-intensity peaks in the centre corresponding to the disordered L2 linker region (Fig. 3b). Other well-dispersed peaks overlap with the spectrum of the isolated PRY/SPRY domain but with reduced intensity (Supplementary Fig. 3b). This is consistent with the previously described weak and transient interaction between CC and PRY/SPRY in solution[35], resulting in an independent tumbling of the CC and PRY/SPRY connected by a flexible linker. Upon addition of pre-let-7 RNA, the PRY/SPRY peaks disappear completely, while the L2 linker peaks remain visible (Fig. 3b). This demonstrates that both domains tumble jointly in the RNA-bound state, which is compatible with two interaction models: i) RNA stabilizes the CC:PRY/SPRY interaction, or ii) RNA is sandwiched by both domains (Fig. 3c).

To further confirm that this conformational change was due to RNA binding, we used solution small-angle X-ray scattering (SAXS). A comparison of the SAXS curves of free TRIM25 CC-PRY/SPRY and bound to pre-let-7 shows a significant decrease in the radius of gyration upon addition of RNA ($R_g = 6.83 \pm 0.05$ nm for the free protein compared to $R_g = 5.7 \pm 0.02$ nm for the complex), which agrees with a more compact conformation than the free protein in which the PRY/SPRY domain is mostly detached from the CC domain in solution[35] (Fig. 3d and Supplementary Table 3). Similar effects were observed for the TRIM25-lnczc3h7a stem-loop complex ($R_g = 5.8 \pm 0.2$ nm) at concentrations well above the $K_D$, confirming that this RNA stem-loop-induced conformation change is the general RNA-binding mechanism of TRIM25 (Supplementary Fig. 3c and Supplementary Table 3). The effect also occurs in SEC-SAXS with an excess of pre-let-7, clearly demonstrating that it is not an artefact caused by the scattering contribution of the free RNA or aggregation (Supplementary Fig. 3d). We conclude that RNA is bound cooperatively and that it enhances the interaction between both domains.

With the aim of generating a TRIM25 mutant with fully deficient RNA-binding activity, we combined the different mutations described in the previous sections (PRY/SPRY-m3, CC-m2 and CC-m4. Figure 3e), resulting in a mutant referred to as CC-PRY/SPRY-m9. ITC and NMR experiments showed that this combined mutant does not bind RNA, while still retaining a folded PRY/SPRY domain and its ability to dimerize via the coiled-coil domain (Fig. 3f and Supplementary Fig. 3e, f). Thus, with a full-length TRIM25-m9 RNA-binding deficient mutant, we established a powerful tool to test RNA binding by TRIM25 in cells and to understand the importance of RNA binding for TRIM25's antiviral activity.

## iCLIP2 reveals TRIM25's RNA structure- and sequence specificity
To investigate the RNA-binding properties of TRIM25 within cells we used HEK293-TRIM25 knock-out (TRIM25 KO) cell lines[17], which contain a flippase recognition target (FRT) site to allow stable integration of a gene of choice into the genome using the Flp-In recombinase[17].

TRIM25-WT, TRIM25-m3 (lacking the RNA-binding properties of the PRY/SPRY domain) and TRIM25-m9 (TRIM25 CC and PRY/SPRY domains RNA binding null) were integrated into the genome, resulting in stable expression at levels close to endogenous TRIM25 in WT cells (Supplementary Fig. 4a). Although TRIM25-m9 retains its E3-ligase activity, it is reduced compared to WT, confirming that RNA binding enhances ubiquitination activity (Supplementary Fig. 4b)[28,29]. To understand the relationship between TRIM25's RNA binding and its antiviral activity, we chose the alphavirus Sindbis (SINV) as model system. SINV is a pathogenic virus transmitted from mosquitoes to vertebrates that is broadly used as prototypal alphavirus. It was chosen for several reasons: firstly, TRIM25 RNA-binding activity is upregulated upon infection with SINV and, secondly, that overexpression of TRIM25 potently reduces SINV infection[43]. In addition, there is no viral protein-dependent suppression of TRIM25 as it occurs with NS1 from influenza A virus[35].

To better understand TRIM25 specificity *in cellulo*, we applied iCLIP2[44] to SINV-infected (referred to hereafter as infected) and uninfected (mock) HA-FLAG-TRIM25 HEK293 cells, as described in Garcia-Moreno et al. [45]. We added a size-matched input (SMI)[46] to correct for background signal. Principal component analysis confirmed the high quality of the iCLIP2 data, showing separated clustering of samples based on sample type (SMI vs IP) and conditions (infected vs mock) (Supplementary Fig. 4c). Strikingly, iCLIP2 revealed that TRIM25 targets significantly more cellular genes and has more binding sites in them after SINV infection (Fig. 4a). In agreement with our biophysical data, TRIM25-m3 showed strong reduction in binding sites and targets, when compared to TRIM25 wild type (WT) protein. Almost no binding sites were observed for TRIM25-m9, suggesting that it also lacks RNA binding *in cellulo*. TRIM25 binding sites in cellular RNAs substantially differ in infected and mock cells, with only ~22% of overlapping targets (Fig. 4b), supporting that TRIM25 RNA-binding activity is regulated by virus infection. TRIM25-m3 binding sites barely overlapped with WT TRIM25, with only 100 (out of 990) being detected by both proteins. This suggests that while TRIM25-m3 is still able to interact with RNA to some extent, its specificity and affinity are substantially affected. The very few binding sites detected in TRIM25-m9 samples did not overlap with neither TRIM25-m3 nor TRIM25 WT (Fig. 4b), suggesting a full or near complete abolishment of its RNA-binding activity to cellular RNAs.

The iCLIP2 results show that TRIM25 interacts predominantly with mRNAs in both SINV-infected and uninfected (mock) conditions (Supplementary Fig. 4d). In uninfected cells, TRIM25 prefers 3'UTRs, followed by coding sequences (CDS), which is similar to a previous study[28]. In SINV-infected cells, this binding preference is reversed, leading to higher binding in CDS than in 3'UTRs. It is important to note that the absolute number of binding sites mapping to the 3'UTR remains similar between the two conditions (Supplementary Fig. 4e). However, most of the new binding sites arising in the infected cells concentrate in CDSs, which changes the distribution across mRNA regions in the metanalysis. Interestingly, these new binding sites emerging in infected cells peak at the start and stop codon, which are key signatures for translational control. One potential explanation is the increased availability of these sequences in infected cells due to low ribosome occupancy as a consequence of the powerful translation initiation shut off occurring after phosphorylation of eIF2α[47]. Interestingly, the TRIM25-m3 binding profile loses the peaks at the start and stop codons and 3'UTR, which indicates that the PRY/SPRY is necessary to define RNA-binding specificity also in infected cells. The very few reads detected in TRIM25-m9 samples were homogeneously distributed across the mRNAs, indicating that they are derived from experimental noise rather than from real TRIM25 binding sites (Fig. 4c).

Since many of the RNAs reported for TRIM25 consist of multiple stem-loops of similar size, we tested whether TRIM25 recognises

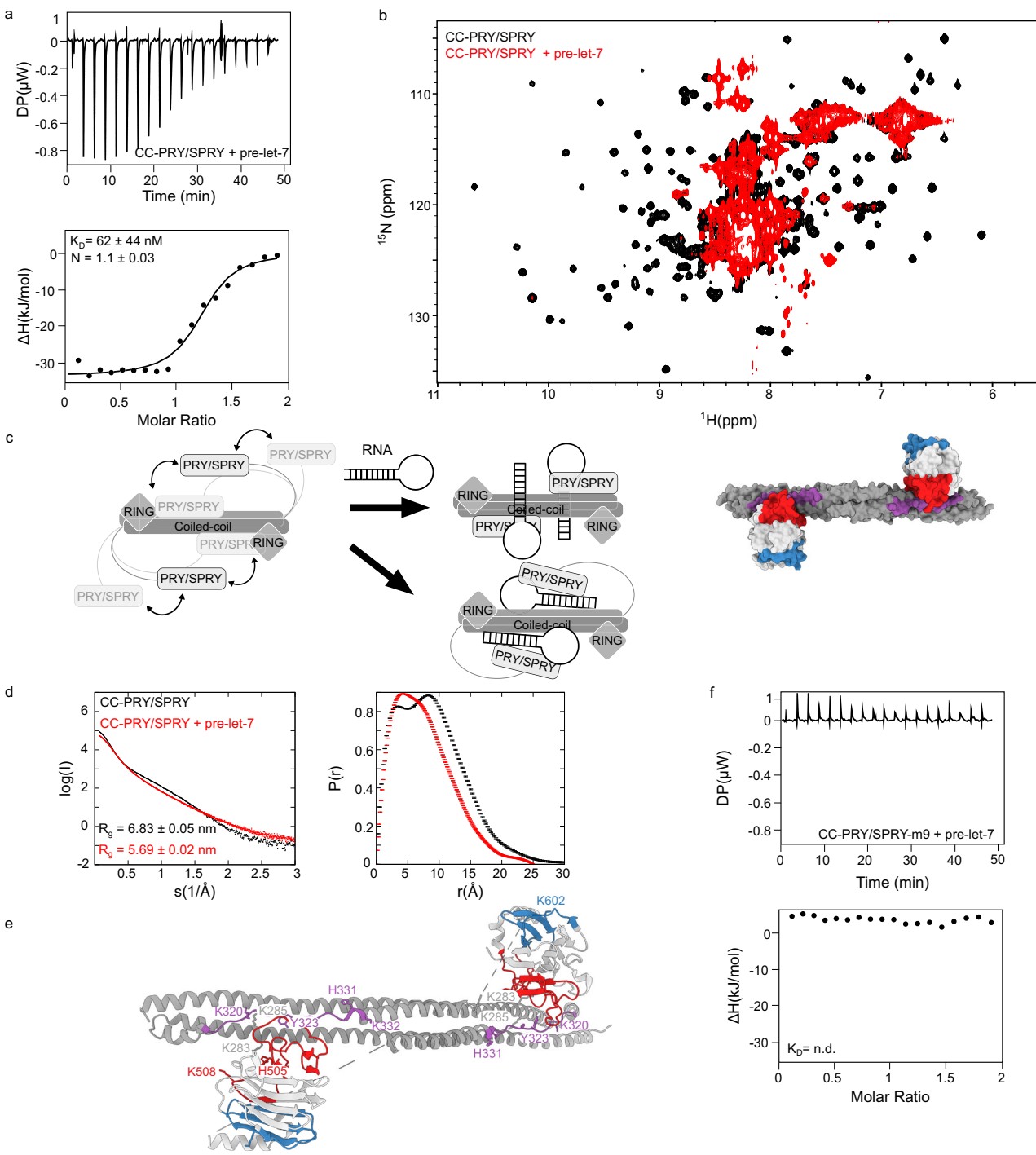

**Fig. 3 | CC and PRY/SPRY domains bind RNA cooperatively. a** Representative ITC binding isotherm for TRIM25 CC-PRY/SPRY (n = 3). The value shown is the average of all replicates and its standard deviation. **b** ${}^1$H,${}^{15}$N-HSQC spectra comparing CC-PRY/SPRY in the absence (black) and presence (red) of pre-let-7. The strong signal loss of peaks corresponding to residues in structured regions of the PRY/SPRY domain upon addition of equimolar amounts of pre-let-7 indicates that the RNA keeps the PRY/SPRY domain at the CC interface leading to joint tumbling and thus increased transverse relaxation and line broadening beyond detection. **c** Proposed mechanisms of RNA-induced conformational change. CC and PRY/SPRY of TRIM25 interact only transiently in the absence of RNA. Binding of stem-loop RNA stabilizes the interaction between the two domains. CC-PRY/SPRY dimer structure (PDB: 6FLN) shown as a surface representation with two binding sites in the PRY/SPRY domain (binding site 1 coloured red and binding site 2 coloured blue) and the binding site in the CC domain (coloured purple). The structure shown was

generated using UCSF ChimeraX[80]. In a second possibility, the stem-loop RNA is sandwiched between the CC and PRY/SPRY domains, which do not interact with each other. **d** SAXS curves and pairwise distance distributions for free TRIM25 CC-PRY/SPRY (black) and its complex with pre-let-7 RNA (red). The distance distribution of the free protein has two maxima, indicating independent tumbling of the CC and PRY/SPRY domains, whereas the distribution for the RNA-bound complex is much narrower and contains only one peak, indicating a conformational change towards a more compact form. **e** CC-PRY/SPRY dimer structure (PDB: 6FLN) in cartoon representation showing all RNA binding sites in the PRY/SPRY domain (binding site 1, red and binding site 2, blue), and CC domain (purple) and residues mutated to obtain the TRIM25 CC-PRY/SPRY m9 mutant. **f** Representative ITC binding isotherm for the TRIM25 CCPRY/SPRY-m9:pre-let-7 complex (n = 3). All experimental setups and ITC measurements including replicates are provided in Supplementary Table 1.

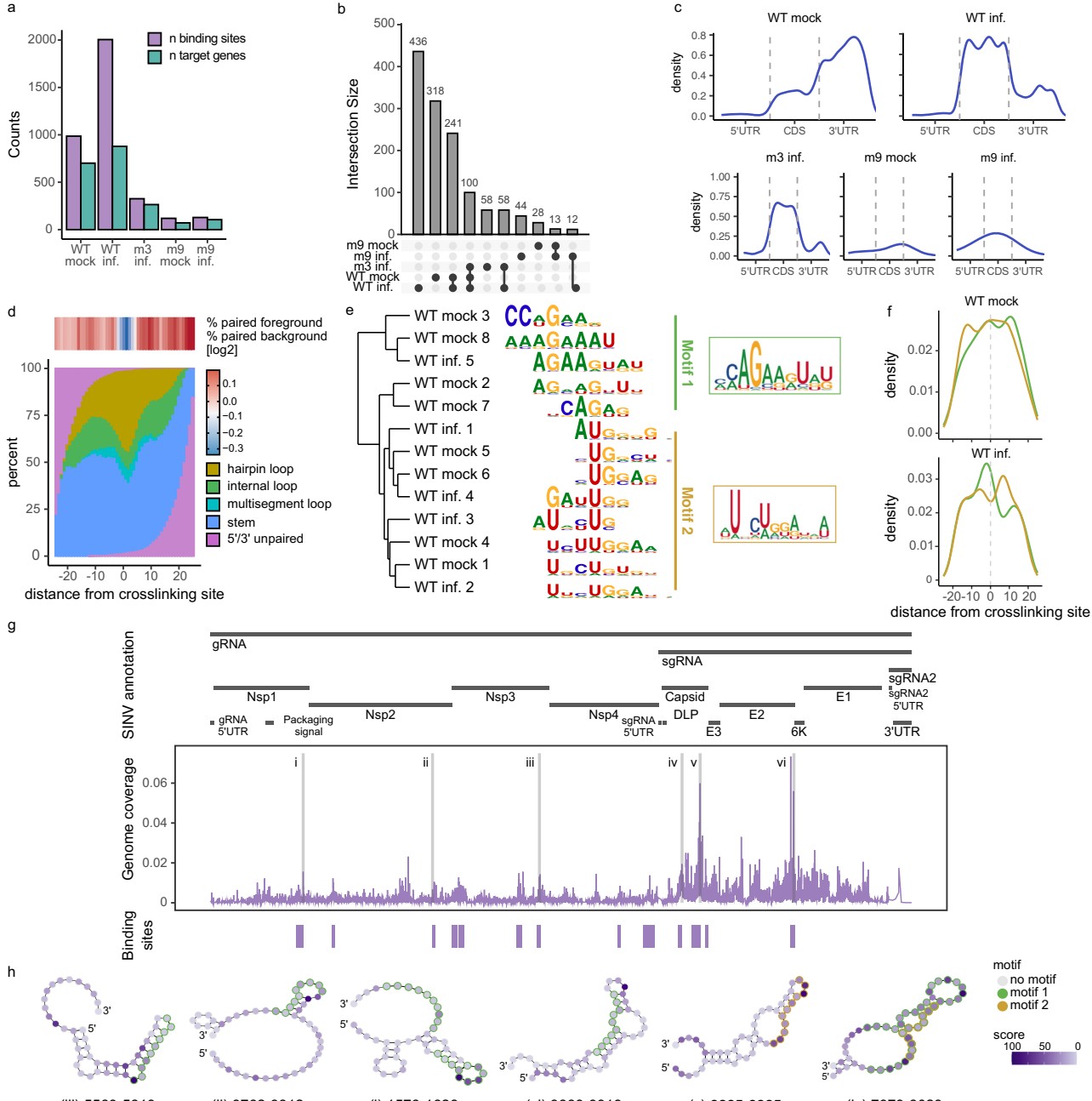

**Fig. 4 | iCLIP2 analysis of TRIM25 reveals structural and sequence signatures in target RNAs.** iCLIP2 was applied to HEK293 TRIM25 KO Flip-In expressing TRIM25-WT or the different TRIM25 mutants (m3 or m9) and infected with SINV and harvested 9 hour-post-infection, hpi (inf.) or mock-infected (mock). **a** Bar plots showing the number of cellular RNA binding sites (purple) and target gene (green) counts identified for the different samples. **b** Intersection plot comparing the different cellular targets RNAs identified in the different samples. **c** Density plot showing the distribution of binding sites across 5' UTRs, CDSs and 3' UTRs on cellular target RNAs for the different samples. **d** Percentage of paired and unpaired sequences across the binding site for the mock WT sample (see Supplementary Fig. 4f for TRIM25-WT infected sample). **e** Motifs enriched in the cellular RNA binding sites of the WT mock and SINV infected samples can be clustered into two prominent classes, an AGAA motif (motif 1) and a UGG motif (motif 2). **f** Density plot showing the distribution of the sequence motifs across the binding site for the TRIM25-WT mock and infected samples (green for motif 1 and orange for motif 2). The dotted line indicates the peak in the crosslinking signal. **g** Binding profile of TRIM25-WT on SINV RNA. Significant binding sites ($p < 0.01$) are indicated as blue boxes underneath the binding site density plot. **h** Secondary structure of the binding sites using SHAPE data[79]. The colour of the nucleotides indicates the crosslinking density at each nucleotide position. The line surrounding the nucleotide indicates the presence of the identified motifs in the RNA structure (green for motif 1 and orange for motif 2).

specific structures by analysing the percentage of predicted base pairing around the crosslink site. Notably, we noticed a low incidence of base pairing at the crosslink site flanked by higher pairing probability in both mock and infected TRIM25-WT samples (Fig. 4d and Supplementary Fig. 4f). This implies that TRIM25 recognises stem-loop structures, with UV crosslinks preferentially in the loop region. To rule

out that this is an artefact of iCLIP2, we extended the analysis to other RBPs, observing that base pairing distributions around the cross-link site are different for other proteins (Supplementary Fig. 4g).

To test whether TRIM25 recognises specific sequences within the stem-loop structure, we searched for enriched sequence motifs for TRIM25-WT within cellular RNAs in both mock and infected samples.

We found two prominent classes (Fig. 4e), an AGAA motif (motif 1) and a UGG motif (motif 2). Detailed analysis of the occurrence of these motifs in the different datasets revealed that in the TRIM25-WT infected sample the AGAA motif typically precedes the crosslinking site, and the UGG motif appears after it. This preference is more pronounced in the infected than in the mock sample probably due to the increase in binding sites upon infection (2032 vs 1019) (Fig. 4f). Of note, these consensus motifs are present in the RNAs used in our biophysical investigation of TRIM25's RNA-binding mechanism (pre-let-7, 28-mer, Lncz3h7a and DENV-SL, Supplementary Fig 4h). We designed a new RNA that was derived from the pre-let-7 but modified to stringently contain both motifs 1 and 2 across the loop region for validation. 1D-$^1$H and 2D-$^1$H/$^1$H-NOESY experiments confirm that this RNA forms base pairs (Supplementary Fig. 4i). The affinity observed for this RNA was lower than for pre-let-7 (180 nM vs 60 nM, Supplementary Table 1, and Supplementary Fig. 4j). This indicates that the inclusion of both motifs does not increase necessarily TRIM25 binding affinity. We also noticed that the shape of the isotherm changes becoming biphasic, as seen previously for other RNAs tested. As expected, the RNA-binding null m9 does not interact with this RNA (Supplementary Fig. 4k).

In addition to binding sites on cellular RNAs, we tested if TRIM25 also binds to the SINV RNA. iCLIP2 revealed several high-probability binding sites across the viral genome. SINV produces two positive sense RNAs, the genomic (g)RNA that encodes the non-structural proteins and is packaged into virions, and the subgenomic (sg)RNA that encodes for the structural proteins. The sgRNA overlaps with the last third of the gRNA and is expressed to a higher level. This explains why the peaks mapping to the sgRNA region are higher in magnitude than those mapping to the first two thirds of the gRNA. Despite this, several binding sites were significantly enriched over the background signal at gRNA and sgRNA regions (Fig. 4g and Supplementary Table 5), suggesting that TRIM25 engages with both viral transcripts. To determine whether these are bona fide TRIM25 binding sites in the SINV genome, we searched for the structure and sequence features that are enriched in the cellular mRNAs (Fig. 4h). To analyse the structural features, we used experimentally determined structural information previously generated by SHAPE reactivity data for the SINV genome[48]. SHAPE reactivity is closely related to nucleotide flexibility, with unconstrained nucleotides showing higher SHAPE reactivity than nucleotides involved in base pairing, stacking or other interactions. RNAfold[49], constrained by the SHAPE data, predicts hairpin loops in most of the binding sites (Fig. 4h). Analysis of the consensus motifs occurrence in the SINV RNA binding sites shows that approximately 70% of the binding sites contain at least one of the motifs (Fig. 4g and Supplementary Table 5).

For the validation of SINV RNA-binding sites obtained by iCLIP2, we designed a new RNA construct called RNA_50_motif1&2 (Supplementary Fig. 5m). This RNA includes nucleotides 1602–1652 of the SINV genome, which overlaps with the TRIM25 SINV binding site 1576–1626 and contains the two identified motifs, with only motif 1 present in the loop. This RNA contains a bulge in the stem similar to DENV-SL (Supplementary Fig. 5n). We confirmed that this RNA features base pairing by 1D-$^1$H and $^1$H/$^1$H-NOESY experiments (Supplementary Fig. 5m, n). Interestingly, we observed the strongest affinity (3.5 nM) for this RNA (Supplementary Fig. 5o). The secondary structure similarity between this RNA and DENV-SL, another tightly bound RNA, suggests that the structure specificity of TRIM25 is more complex than merely binding to single stem-loop structures containing motifs 1 & 2. This confirms the iCLIP2 results showing that TRIM25 can bind viral RNA motifs with high affinity in vitro.

### RNA binding is critical for TRIM25's antiviral activity

TRIM25 is known to inhibit viral replication, as seen for Dengue, Influenza A and Rabies viruses[29,50]. However, whether its RNA-binding activity is important for this antiviral role remains poorly understood. To test this, we infected TRIM25 KO cells rescued with either TRIM25 WT, mutants affecting either RNA-binding (m3, m9) or the E3 ligase activity (E3-dead[18]) with SINV.

We analysed the effect of our RNA-binding mutants on viral fitness using chimeric viruses expressing mScarlet (SINV$_{NSP3-mScarlet}$) or mCherry (SINV$_{mCherry}$) from the genomic or a duplicated subgenomic RNA, respectively (Supplementary Fig. 5a). Rescue of the KO line with WT TRIM25 decreased SINV fitness with similar strength to the parental line, validating the assay (Supplementary Fig. 5b). Notably, we observed that mutants m3 and m9 increased viral fitness relative to TRIM25-WT to a similar extent than the E3-dead mutant (Fig. 5a). These results suggest that both the E3 ligase activity and the RNA-binding activity are equally important for TRIM25's antiviral activity against SINV. Red fluorescence derived from SINV$_{NSP3-mScarlet}$ and SINV$_{mCherry}$ is a proxy for the translation of the gRNA and sgRNA, respectively. TRIM25 suppressed the red fluorescence protein expression from both gRNA and sgRNA, which is compatible with the presence of TRIM25 binding sites in both RNAs (Fig. 5a). We orthogonally validated these experiments using Western blot to quantify capsid levels (Supplementary Fig. 5c). Again, the KO cell line rescued with WT TRIM25 restores the antiviral function to the levels of the parental cell line (Supplementary Fig. 5d). It is important to stress that the increase in capsid observed between the TRIM25-WT and the different mutants is significant in both fluorescence-based and Western blot approaches. However, the differences between m9, m3 and E3-null are non-significant although there is a trend for m9 to cause the strongest effects (Fig. 5a and Supplementary Fig. 5c).

SINV replicates in invaginations in endosome-derived membranes, referred to as replication organelles (ROs), where the replicating viral RNA is isolated from the host cytosol[51–53]. TRIM25 is cytosolic in uninfected cells, but it accumulates in ROs upon SINV infection, together with viral RNA and capsid protein[43] (Fig. 5b). To determine whether the ability of TRIM25 to interact with SINV RNA promotes its redistribution to ROs, we used immunofluorescence and RNA single molecule in situ hybridisation (smFISH). TRIM25-WT and TRIM25-m9 showed diffuse localization in the cytoplasm of uninfected cells. Upon infection, TRIM25-WT accumulated into cytoplasmic foci that co-localise with the viral RNA and are consistent with ROs (Fig. 5b). Conversely, TRIM25-m9 remained diffuse after infection and did not accumulate in these foci (Fig. 5b). These results indicate that the RNA-binding activity of TRIM25 is important for its localization to ROs. We hypothesise that interaction with viral RNA can both recruit TRIM25 to the ROs and potentially regulate its E3 ubiquitin ligase activity as previously proposed[17].

## Discussion

Several attempts have been made to describe the RNA-binding mechanism of TRIM25[28,29], however a complete description of how TRIM25 recognises the RNA, and its specific motif is still lacking. We have been able to identify all the RNA binding regions of TRIM25 at residue resolution, revealing the cooperative behaviour between the PRY/SPRY and CC domains, and describe a consensus RNA motif by combining a variety of biophysical methods (NMR, ITC, SAXS, SEC-MALS) and iCLIP2. Our results show that TRIM25 achieves very tight binding of RNA by using multiple sites on the CC and PRY/SPRY domains, with each binding site contributing cooperatively to affinity and RNA specificity. Not only were we able to precisely describe the residues involved in RNA binding, but we also discover that the PRY/SPRY domain has a dual binding mode with one site having single-stranded RNA specificity and the other site preferring double-stranded RNA. The proximity of these two sites suggests structural specificity, and indeed we found that the binding of the CC-PRY/SPRY domains to several stem-loops is within nanomolar affinity. This extra layer of complexity may explain the failure of previous studies at identifying a

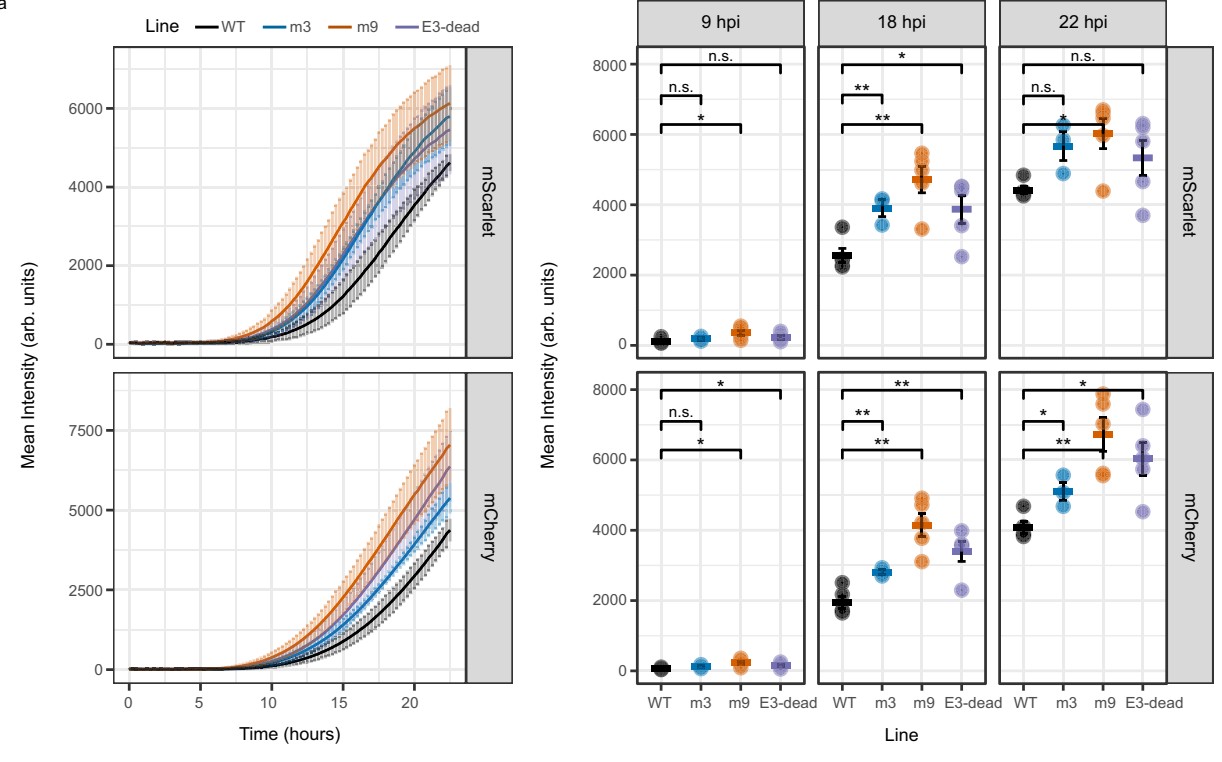

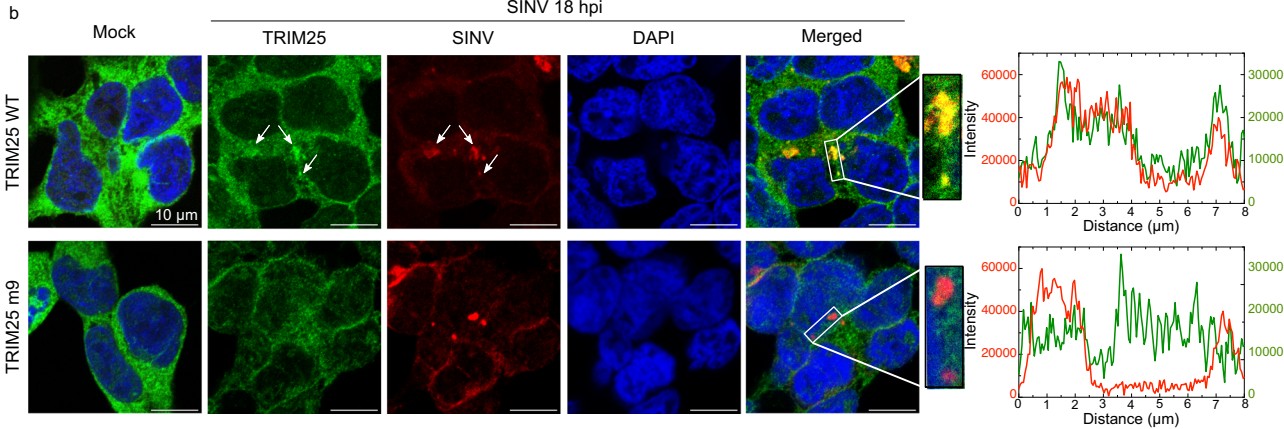

**Fig. 5 | Impact of the RNA binding activity of TRIM25 on its antiviral activity.** **a** Red fluorescence signal in TRIM25 WT (black) and mutant lines (m3 in blue, m9 in orange and E3-dead in purple) infected with SINV-mCherry (left panel) or SINV-nsp3-mScarlet (right panel). Fluorescence was measured every 15 min in a plate reader with atmospheric control (5% $CO_2$ and 37 °C). The fluorescence is represented as mean ± SD of six independent infections in three biological replicates.

Statistical differences are based on a two-tailed homoscedastic t-test (***$p < 0.001$; **$p < 0.01$; *$p < 0.05$). No adjustments were made for multiple comparisons. **b** Localization analysis by immunofluorescence and smFISH of TRIM25 (green) and SINV RNA (red). Nuclei are labelled with DAPI. Green and red fluorescence profiles for regions of interest are displayed on the right.

distinct target sequence motif for TRIM25. Here, we show that TRIM25 is indeed a bona fide and highly specific RBP.

The molecular dissection of TRIM25's RNA-binding activity allowed us to generate a mutant that does not bind RNA neither in vitro nor in cells, without affecting its fold or dimerization property. With this tool in hand, we managed to elucidate the RNA-binding specificity of TRIM25 inside the cell.

Previous iCLIP data on TRIM25[17,28] showed that TRIM25 prefers G- and C-rich sequences, but no strong consensus motif was identified. Our iCLIP2 data allowed us to describe a structural preference of TRIM25 for stem-loop structures inside the cell, including both host and viral RNA. This observation can also explain the G- and C-rich

sequences observed in the previous dataset, as G-C base pairing is more stable than A-U due to the higher number of hydrogen bonds formed. We also identified two preferred sequence motifs within the hairpin loop of these stem-loop structures. Particularly, we found that the motif sequences have the common denominator of being A/G rich which also happens for the binding sites for SINV-RNA that do not have any of the motifs present. Interestingly, the sequence motifs found are both present in previously identified RNAs that bind TRIM25, such as pre-let-7 RNA and DENV-SL (Supplementary Fig. 4g). Importantly, the iCLIP2 data showed the lack of RNA-binding activity of TRIM25-m9, which is an important confirmation in cells of the mutations inferred by our biophysical data.

An open question is how TRIM25 exerts its antiviral activity and whether RNA binding is involved. There have been some efforts to answer these questions[28,29], but these studies used TRIM25 constructs that retained RNA-binding activity to some extent (Supplementary Fig. 3). Recently, it was shown to bind and destabilize IAV mRNAs[17]. Our data provide several lines of evidence supporting that RNA binding is an essential feature for the antiviral activity of TRIM25. We have shown that viral gene expression is strongly enhanced when RNA binding is completely abolished, phenocopying the mutant that lacks the E3 ligase function. In addition, our iCLIP2 data showed that TRIM25 binds to specific regions of viral RNA. Indeed, our microscopy analyses revealed that TRIM25 RNA-binding activity is a prerequisite to drive the localisation of TRIM25 to SINV ROs, which is likely a critical step for triggering its antiviral activity. This activity could be direct or through interaction with ZAP, which is known to localise to ROs upon SINV infection[54]. Understanding whether TRIM25 RNA-binding activity is important for ZAP localisation, or whether interaction of TRIM25 with ZAP is essential for the antiviral effect, as well as exploring how TRIM25 condensates within ROs, will contribute to a better understanding of the antiviral mechanism driven by TRIM25.

## Methods

### Protein expression and purification
TRIM25 CC (aa 189–379), PRY/SPRY (439–630) and CC-PRY/SPRY (189–630) were cloned using restriction free cloning into pETM22 featuring a 3C-protease cleavable hexa-histidine and thioredoxin tag (His-Trx). N-terminally extended PRY/SPRY construct (407–630) was cloned into pETM20 with a TEV-protease cleavable His-Trx tag. TRIM25 189–379 was expressed in BL21(DE3). TRIM25 439–630, 407–630 and 189–630 were co-expressed with chaperones KJE, ClpB and GroELS in *E. coli* BL21(DE3)[55]. All proteins were induced at OD600 = 0.6 by 200 μM isopropyl-ß-D-1-thiogalactopyranoside (IPTG, ROTH CN08.2) and incubated at 18 °C for 20 h.

All proteins were purified by immobilized Nickel affinity chromatography (Histrap HP, Cytiva 17524802) in 50 mM Tris, pH 7.5, 300 mM NaCl and 0.2 mM TCEP (ROTHHN95.2) and eluted with a gradient of imidazole (10–300 mM). In the case of co-expression with chaperones the column was washed with 50 mM Tris, pH 7.5, 350 mM KCl, 5 mM MgSO4 and 1 mM ATP(Merck, A9187) prior to elution. The tag was removed by 3C- or TEV-protease digestion overnight and an additional passage over the Histrap HP column (CC and extended PRY/SPRY) or GE Healthcare HiTrap SP HP cation exchange column (Cytiva 17115201, PRY/SPRY, residues 439–630 and CC-PRY/SPRY, residues 189–630). As a final step, aggregates were removed by gel filtration on a GE Healthcare Superdex S75 (Cytiva 29148721, CC, PRY/SPRY, extended PRY/SPRY) or S200 (Cytiva 28990944, CC-PRY/SPRY) in 20 mM MES (ROTH 4259.5), pH 6.5, 75 mM NaCl and 0.5 mM TCEP. For stable isotope labelling, proteins were expressed in M9 medium using $^{15}NH_4Cl$ (Sigma299251, 0.5 g/l) as sole nitrogen and carbon source.

### RNA production
Pre-let-7a-1@2 (5′-GUA UAG UUU AAA AGG AGA UAA CUA UAC −3′), Lnczc3h7a-SL (5′-UUUUAUCUGAGUUGGAGGUGAAG-3′) and DENV-SL (5′- GCA GGU CGG AUU AAG CCA UAG UAC GGG AAA AAC UAU GCU ACC UG-3′) were in vitro transcribed from DNA oligos using T7 RNA polymerase. Reaction mixtures containing 2 μM forward and reverse primer, 40 mM Tris, pH 8.0, 0.2 mM MgCl₂, 10 mM NTPs (NEBN0446S), 10 mM spermidine, 15 mM DTT, 0.01 % Triton X-100, 4 U/ml TIPP, 0.1 mg/ml T7 polymerase were incubated for 5 h at 37 °C and extracted by chloroform/phenol treatment. The aqueous phase was further purified by preparative gel electrophoresis via denaturing PAGE or HPLC using a Thermo DNA Pac PA100 22 x 250 mm anion exchange column at 95 °C. The purified RNA was dialyzed against 20 mM MES, pH 6.5, 75 mM NaCl and 0.5 mM TCEP and refolded before use by heating up at 95 °C for 5 min and snap-cooling on ice. The shorter pre-let-7 loop

(5′-UAA AAG GAG AU-3′), stem-fusion (5′-G UAU AGU U C AAC UAU AC-3′), 28-mer duplex RNA (5′-AUG GCU AGC UGG AGC CAC CCG CAG UUC G-3′), pre-let-7a-1@2, pre-let-7_modified (5′- GUA UAG UUU AAA GAA UGG AUA ACU AUA C-3′) and RNA_50_motif1&2 (5′- UUA GUG GCA GAC AAA GGC AUC GAG GCA GCC GCA GAA GUU GUC UGC GAA GU-3′) constructs were purchased from IBA or Biomers.

### Nuclear magnetic resonance spectroscopy
NMR spectra were acquired on Bruker Avance III spectrometers operating at magnetic field strengths corresponding to proton Larmor frequencies of 600, 700, and 800 MHz, equipped with cryogenic triple resonance probes (600 and 800 MHz) and a room temperature triple resonance probe (700 MHz). Protein observed experiments were performed in 20 mM sodium phosphate, pH 6.5, 150 mM NaCl, 2 mM TCEP, 5% $D_2O$ and 0.02% sodium azide at 293 K. For RNA observed experiments samples were prepared in 20 mM MES, pH 6.5, 75 mM NaCl, 5 mM $MgCl_2$, 0.02% sodium azide and measured at 278 K.

For titrations 100 μM $^{15}N$-labelled TRIM25-PRY/SPRY (439–630) was titrated with stock solutions of 10–20 mM natural abundance RNA to a molar excess of 2.5–3 (details indicated in the Fig. 1b). At each titration point a $^{1}H,^{15}N$-HSQC spectrum was acquired. Due to protein precipitation, a different number of scans were used for different titration points (256 scans for PRY/SPRY domain alone, 512 scans for 1:1 and 1:2 and 1024 for 1:3). Figure 1B was generated using the same number of contour and noise thresholds. Spectra were processed using NMRPipe[56], visualized and peak shifts tracked using SPARKY[57]. Peak shifts for PRY/SPRY 439-630 were assigned based on a previously published assignment[35]. CSPs in histograms (Fig. 1d) were determined at a protein:RNA ratio of 1:3.

### Isothermal titration calorimetry
Isothermal titration calorimetry (ITC) data was collected on a Malvern MicroCal PEAQ-ITC at 20 °C in 20 mM MES, pH 6.5, 75 mM NaCl and 0.5 mM TCEP. Depending on the protein construct, RNA was titrated from the syringe at concentrations ranging from 20 to 1100 μM into the cell containing protein with concentrations between 2 and 150 μM while stirring at 750 rpm (see Figs. 1, 2 and 3 and Supplementary Figs. 1, 2 and 3). For experiments with DENV-SL the RNA was kept in the cell at 15 μM and the protein titrated from the syringe at 110 μM. Experiments were typically done in triplicates and analysed using the Malvern MicroCal PEAQ-ITC analysis software. Data was fitted by assuming the simplest binding mode necessary to fully explain the data (one or two binding sites).

### Sequence alignments
To reveal the conservation of TRIM25-PRYSPRY among TRIM proteins, sequences of the 44 human TRIM proteins belonging to group IV (containing the PRY/SPRY domain) and Riplet (RN135) were obtained from InterPro[58] using the PRY/SPRY domain as a seed. The alignment of the different PRY/SPRY domains was performed using ClustalW[59] and the alignment is provided in Supplementary Fig. 1g. Using the 44 sequences of human TRIM proteins containing the PRY/SPRY domain used previously, the structure of the proteins was modelled using Alphafold[60,61] to determine the boundaries of the CC domain for each protein. The alignment of the different CC domains was then performed using ClustalW [59] and shown in Supplementary Fig. 2f.

To test sequence conservation of TRIM25 amongst different species, sequences of 18 TRIM25 proteins from different species were aligned using ClustalW[59] and the full alignment for the PRYSPRY domain is shown in Supplementary Fig. 1h. Sequences were obtained from ENSEMBL[62] using the human TRIM25 protein as seed (UniProt ID: Q14258). Details of the species are given in Supplementary Data Table 2. In addition, the sequence logo corresponding to the CC specific region responsible for RNA binding was generated by WebLogo (http://weblogo.berkeley.edu/logo.cgi)[63] using the aligned protein

sequences corresponding to the CC domain from the same 20 species (Supporting Table 2 and Supplementary Fig. 2f). The height of the stack of letters at each position reflects sequence conservation and reaches a maximum value of 4.32 for proteins. The height of each letter within a stack is proportional to its abundance.

## Small-angle X-ray scattering

SAXS data were collected at the beamline P12, operated by EMBL Hamburg at the PETRA III storage ring (DESY, Hamburg, Germany)[64]. Measurements were done at 20 °C in 20 mM MES, pH 6.5, 75 mM NaCl and 0.5 mM TCEP in flow cell mode at 1.24 Å. For each sample and buffer 20–100 frames with 0.05–0.195 s exposure time were acquired and frames showing radiation damage manually removed. SEC-SAXS data was collected at the BM29 BioSAXS beamline using a wavelength of 0.99 Å and a Superdex 10/300 S200 Increase size exclusion column and analysed using Chromixs[65,66]. Per run 2500 frames with 1 s exposure time were collected. Data analysis was done using the ATSAS package version 2.8.3[67]. PRIMUS was used for frame averaging and buffer subtraction[68]. The radius of gyration, $R_g$, was estimated using the Guinier approximation in PRIMUS. Pair-wise distribution functions were calculated using GNOM[69]. Collection statistics for SAXS measurements are summarized in Supplementary Table 3.

## Circular dichroism spectroscopy

CD spectra were acquired using a Jasco 815 circular dichroism spectrometer. Samples were measured at 20 °C in quartz cuvettes with 1 mm path length in 20 mM MES, pH 6.5, 75 mM NaCl and 0.5 mM TCEP at 0.5 mg/ml.

## Protein-RNA crosslink identification by CLIR-MS/MS

TRIM25 crosslinks were identified by CLIR-MS/MS[39]. Experiments were performed using a Thermo Orbitrap Fusion Lumos mass spectrometer with settings as described in Vries et al.[70] and data analysis was performed using RNxQuest, version 1.0[71]. As shown in Fig. 2e, the cross-linked peptide is detected as a triply charged ion with a mass difference of approximately 11 Da, corresponding to the introduction of $^{13}C$ and $^{15}N$ stable isotopes in the heavy RNA. MS/MS spectra of the peptide crosslinked to light U (right, top) or heavy U (right, bottom). Assigned fragment ions with a relative intensity greater than 5% are labelled. All assigned fragment ions (b and y ions) are shown on the peptide sequence in the lower right-hand corner. The localisation of the crosslinking site to the tyrosine residue (highlighted in bold) is confirmed by observing the unmodified b7 ion and the b8 ion carrying the U modification. All fragment ions carrying the U modification (e.g., $b_8$, $b_9$, $y_{15}$) show a mass shift between the light and heavy form spectra.

## Size exclusion chromatography – multi-angle light scattering (SEC-MALS)

50 μl of CC domain or the mutants were injected onto a Superdex 200 5/150 GL gel-filtration column (Cytiva 28990945) in 20 mM MES, pH 6.5, 75 mM NaCl and 0.5 mM TCEP, at a flow rate of 0.3 ml/min and at room temperature. The column was connected to a MiniDAWN MALS detector and Optilab differential refractive index detector (Wyatt Technology). Data were analyzed using the Astra 7 software (Wyatt Technology).

## Plasmids and recombinant DNA procedures

Synthetic genes for generation of inducible cell lines were obtain from Genscript. cloned into the pcDNA5/FRT with the HA-FLAG tag followed by a flexible linker encoding for GGSGGSGG (glycine and serine repeats) to facilitate the folding of the RBP of interest independently from the tag. The E3-dead mutant was generated by replacing Cys50 and Cys53 with serines which abolishes the E3 ligase activity of the RING domain as shown by Zheng et al.[18].

## Cell lines generation

To generate the HEK293 TRIM25-WT, TRIM25-m3, TRIM25-m9 and TRIM25- E3-ligase dead cell lines, HEK293 TRIM25 KO cells were transfected with pOG44 and the corresponding pcDNA5/FRT plasmid containing the protein of interest using X-tremeGENE 360 transfection reagent following manufacturer's recommendations (Roche, 8724121001). For the selection of inducible cell lines, 50 μg/ml Hygromycin B was added to the media. Cells were selected and checked TRIM25 variant expression by western blotting (Supplementary Fig. 4a).

## In vivo ubiquitination assays

Briefly, HEK-293 TRIM25 KO cells were seeded the day before transfection. Next day, equal amount of Ub and HA-tagged TRIM25 constructs (WT, m9 and E3-dead) plasmids were mixed with Fugene HD transfection reagent (Promega no. E2311) in reduced serum medium. The transfection mix was incubated at room temperature for 5 min before adding dropwise to the cells. The transfected cells were incubated overnight at 37 °C, 5%CO₂. Twenty-four hours post-transfection, the proteasome was inhibited with a final concentration of 0.5 μM Carfilzomib/PR-171 (Selleckchem no. S2853) and put back in the incubator. After 6 h the cells were harvested.

The harvested cells were lysed in 1x cell lysis buffer (50 mM Tris-HCl pH 7.5, 150 mM NaCl, 0.5% IGEPAL (NP-40) and 50 mM NEM, cOmplete protease inhibitor cocktail (Sigma Aldrich)) for 30 min at 4 °C. Then the lysate was spin down for three minutes at 1500 rpm to sediment nuclei, transferred to a fresh tube and then spin down for 10 min at 17,000 x g at 4 °C. After pelleting the debris the protein concentrations in the supernatant were measured using Qubit Protein Broad Range Assay Kit (Q33211). The protein ubiquitination was analyzed by western blot with anti-TRIM25 (Supplementary Fig. 4b).

## Western blotting

Samples were prepared by mixing with NuPAGE LDS Sample Buffer 4X (Thermo Fisher Scientific, #NP0008) and incubated at 95 °C for 5 min. Proteins were resolved by SDS-PAGE for 50 min at 180 V and transferred to nitrocellulose membranes using the Trans-Blot Turbo Transfer System (Bio-Rad). Membranes were blocked for 1 hour at RT or overnight at 4 °C. 5% skimmed milk in 0.1% PBS-T was used for blocking and for the preparation of antibody dilutions. Membranes were incubated with primary antibodies either for 1 h at RT or overnight at 4 °C. After three washes in 0.1% PBS-T, the membranes were incubated with the appropriate secondary antibodies for 1 h at RT. The membranes were then washed three more times in 0.1% PBS-T prior to imaging. LI-COR Odyssey CLx imaging system for visualisation and Image Studio software for quantification.

## Viruses

Sindbis virus was produced by in vitro transcription from the plasmid pT7svwt[72]. SINV-mCherry and SINV-nsp3-scarlet was generated by duplication of subgenomic promoter and insertion of mCherry[43] (Supplementary Fig. 5a). SINV-nsp3-scarlet was generated by amplifying mScarlet sequence from pUC19_CRISPR_CT_Scarlet_GSG_P2A_BSD (primer forward: GGGGACTAGTATGGTGAGCAAGG; reverse: ccccAC TAGTCTTGTACAGCTCG) and cloned into pT7SINV-WT using SpeI restriction enzyme (Supplementary Fig. 5a).

## Fluorescence-based viral fitness assay

$6 \times 10^4$ cells were seeded on each well of a 96-well microplate with flat μClear bottom (Greiner Bio-One, #655986) in DMEM lacking phenol-red (Gibco # 31053028) and supplemented with 2% FBS and 1 mM sodium pyruvate. The different cell lines (KO rescued WT, E3-ligase dead, m3, and m9) were infected with SINV-mCherry or SINV-nsp3-mScarlet (see Supplementary Fig. 5a) at 0.1 MOI in complete DMEM lacking phenol-red and supplemented with FBS 2%. Cells were

incubated at 37 °C and 5% $CO_2$ in a CLARIOstar fluorescence plate reader (BMG Labtech) for 24 h in FBS 2%; mCherry signal was monitored by measuring fluorescence (mCherry: excitation 570 nm, emission 620 nm; mScarlet: excitation 569 nm, emission 594 nm) every 15 min. Statistical significance of the difference in mCherry or mScarlet expression at 18 and 22 hpi was determined by t test (from n > 3).

## iCLIP2

To identify TRIM25 binding sites for RNA on mock and SINV-infected cells at a high resolution, we employed iCLIP2-seq[44] with the modifications explained in Garcia-Moreno et al. [45]. $10 \times 10^6$ HEK293 TRIM25 WT, TRIM25-m3, and TRIM25-m9 cells were seeded in 6 sets of 10 cm dishes and incubated for 24 h. Three cell sets per cell line were then infected with 3 MOI of SINV. At 9 hpi, cells were washed with PBS 1x, and UV irradiated with 300 mJ/cm² at 254 nm. To obtain RNA fragments of suitable length and to degrade DNA, 1 ml of thawed lysate of each replicate was incubated with 10 U Ambion RNase I (Thermo-Scientific, #AM2295) and 4 U Turbo DNase per ml of lysate for 3 min at 37 °C, with 1100 rpm agitation. Subsequently, lysates were placed on ice and supplemented with 200 U RiboLock RNase Inhibitor (ThermoScientific, #EO0382). Then, 10 µl of lysate was taken from each sample for the size-matched input (SMI). 30 µl of ANTI-FLAG M2 Magnetic Beads (Millipore, #M8823) slurry per ml of lysate was pre-equilibrated in lysis buffer and resuspended in 50 µl of lysis buffer. Beads were added to the lysate and incubated for 2 h at 4 °C with gentle rotation. Next, we followed the protocol, explained in Garcia-Moreno et al. [45]. To decide the number of cycles, a qPCR was performed using EvaGreen (Biotium, #31000), 2X Phusion HF PCR Master mix and P5/P3 Solexa primers (the optimal cycle number was considered as the cycle number at the centre at the "knee of exponential growth" −3). The number of cycles used for each replicate can be found in the supporting information (Supplementary Table 4). The final PCR products were purified by using ProNex beads (Promega, # NG2001) which allow for selecting the DNA of the correct size. The libraries were quantified using Qubit and the size was determined by using High Sensitivity D1000 TapeStation. Each group of samples was pooled equimolarly and then mixed considering the number of samples of the IP and the SMI and the ratio of reads. The final library was processed using single-end sequencing mode with a NextSeq 500/550 High Output v2 kit (75 cycles, Illumina).

### Data processing for iCLIP2 and statistical analysis

Raw FASTQ files were demultiplexed using the Je Suite[73] and adapters were trimmed using Cutadapt[74]. STAR was used to align reads to a concatenated human (GRCh38, ENSEMBL Release 106) and SINV (pT7-SVwt) genome in end-to-end alignment mode[75]. Only uniquely aligned reads were retained for downstream analysis. PCR duplicates were collapsed using unique molecular identifiers (UMIs) with the Je Suite. The crosslink truncation site for each read (−1 from the 5′ start site of the read) was extracted using BEDTools[76]. Peak calling was performed with HTSeq-clip and the R/Bioconductor package, DEW-seq[77]. HTSeq-clip was used to generate a sliding window annotation of the human and SINV genome (50nt window, 20nt step size) and calculate the frequency of crosslink truncation sites within each window. DEW-Seq was then used to calculate the differential enrichment of each window relative to size matched input control samples, with a cut-off of 2 log2 fold change and 0.01 adjusted p-value. Multiple hypothesis correction was performed using the Independent Hypothesis Weighting (IHW) method[78]. Overlapping windows were merged to form binding regions. PCA analysis was performed using DESeq2. Following size correction and variance stabilisation, the 1000 most variable sliding windows were selected and used for PCA plotting. Binding site properties, including gene name, biotype, gene feature, were extracted from the ENSEMBL genome annotation using the GenomicRanges package. Metagene analyses were performed using

functions from the cliProfiler package. Sequences for motif prediction and secondary structure prediction were defined for each binding site as a 50-nucleotide region, centred on the peak in BigWig signal. They were extracted from the CDS sequence of the longest isoform of the gene in which the binding site occurred. For motif prediction, a gene and gene region-matched background sequence was extracted for each binding site to allow for differential enrichment analysis. Enrichment analysis was performed using HOMER. MotifStack was used to cluster and plot motifs. Secondary structures were predicted using RNAfold from ViennaRNA[49] and specific structure features were extracted by forgi. Enrichment for paired sequence was calculated relative to 10 scrambled controls of each input sequence. For plotting of SINV viral genome structures, SHAPE data from Kutchko et al. [79] was included as a constraint in the structure prediction. Coordinates for plotting specific structures were generated with the RRNA package. SINV genome coverage in reads per million was calculated using BEDTools. SMI signal was subtracted from IP signal for plotting.

### Clustering analysis of the percentage of paired and unpaired sequences

To determine the optimal number of clusters for the data, the R package NbClust was used. This package compiles several different methods for identifying the optimal number of clusters. After running the NbClust function, the number of clusters most frequently identified as optimal was either 2 or 5. To provide a more comprehensive overview of the data 5 clusters were used. Subsequently, the K-means clustering was performed using the FKM function from the fclust package in R. This method allowed us to classify the sequences into 5 distinct clusters. For each cluster, we calculated the mean ratio of paired to unpaired sequences at each position. This approach enabled us to analyse and interpret the distribution and characteristics of paired and unpaired sequences within each cluster effectively.

### Combined inmunofluorescence and RNA smFISH assays

$1.8 \times 10^5$ cells were seeded on High Precision Coverslips (Merienfeld, #0117520) and incubated for 24 h in DMEM with 10% FBS. Cells were either mock-infected or infected with 1 MOI of SINV in DMEM without FBS at 37 °C, followed by the replacement of the medium with DMEM supplemented with 2% FBS. At 18 h post-infection, cells were washed with PBS and fixed in 4% paraformaldehyde for 10 min at room temperature. After three 5 min washes in PBS, the cells were permeabilised with 1x PBS supplemented with 0.1% Triton X-100 for 10 min at room temperature. Cells were washed twice with 1X PBS for 5 min each, twice with 2x SSC for 5 min each, and twice with pre-warmed pre-hybridization buffer (2x SSC and 10% deionized formamide in DEPC-treated water) at 37 °C for 20 min each. The cells were then incubated with 125 nM SINV RNAs specific Stellaris probes (LGC Biosearch Technologies) in pre-heated hybridization buffer (2x SSC, 10% deionized formamide and 10% dextran sulphate in DEPC water) at 37 °C in a humidified chamber overnight. Cells were then washed twice with pre-warmed pre-hybridization buffer for 20 min at 37 °C, twice with 2X SSC for 5 min each, and twice with 1X PBS 5 min each. All solutions used from now on were RNase free by adding RiboLock RNase Inhibitor (Thermo Fisher Scientific, # EO0381). Cells were incubated with 1X PBS-BSA 5% for 30 min followed by primary antibody (HA antibody, Biolegend 901502 1:250 dilution) in 1X PBS-Tween20 0.1% (PBST) supplemented with 5% BSA for 1 h. Next, the cells were washed three times with PBST for 5 min each and were then incubated with secondary antibody (α-rabbit Alexa488 at 1:500 dilution) in 0.1% PBST supplemented with 5% BSA for 1 h in the dark. The cells were washed three times in 0.1% PBST and once in 1X PBS for 5 min each. Cells were then incubated with DAPI (1:500 of 0.5 mg/mL stock) in 1X PBS for 10 min at room temperature, followed by two washes with 1x PBS for

5 min and one wash with DEPC-treated pure water for 1 min. Finally, coverslips were collected and mounted on glass slides using ProLong Diamond Antifade Mounting Medium (Catalog number: P36961).

## Image acquisition
Cells were imaged on a Zeiss LSM880 confocal system using a 63X oil objective (Plan-Apochromat 63x/1.4 Oil DIC M27). Images were generated using Zeiss Zen 3.4 Blue Edition software.

## Reporting summary
Further information on research design is available in the Nature Portfolio Reporting Summary linked to this article.

## Data availability
The data supporting the findings of this study are available from the corresponding authors upon request. The crystal structures of human TRIM25 PRYSPRY, CC and CC-PRY/SPRY are available in the PDB under the accession numbers 6FLM, 4LTB and 6FLN. SAXS data has been deposited in the SASBDB under accession codes SASDK78, SASDK88 SASDK98, and SASDKA8. The iCLIP2 data is available under accession code GSE276465. Source data are provided with this paper.

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

## Acknowledgements

We thank the EMBL Genomics Facility, especially Ferris Jung, for the data acquisition. We thank the EMBL Data Science Internal Support team, especially Sarah Kaspar, for the statistical consulting. L.A. was supported by a fellowship from the EMBL Interdisciplinary (EI4POD) program under Marie Skłodowska-Curie Actions COFUND (847543) and acknowledges support by the Joachim Herz Foundation through an Add-on Fellowship for Interdisciplinary Life Sciences. L.I. is funded by BBSRC DTP scholarship number BB/M011224/1 V.R. is funded by the European Union's Horizon 2020 research and innovation programme under Marie-Sklodowska-Curie n. 892756. K.R. is supported by the Francis Crick Institute, which receives its core funding from Cancer Research UK (CC2075), the UK Medical Research Council (CC2075), and the Wellcome Trust (CC2075). A.L. acknowledges Paola Picotti (ETH Zurich) for access to instrumentation and laboratory infrastructure. G.M. is funded by project financed under Dioscuri, a programme initiated by the Max Planck Society, jointly managed with the National Science Centre in Poland, and mutually funded by Polish Ministry of Science and Higher Education and German Federal Ministry of Education and Research (2019/02/H/NZ1/00002). G.M. was also a recipient BBSRC project grant (BB/T002751/1). F.A. acknowledge support of the SNSF NCCR RNA and Disease (51NF40-182880). A.C. is funded by the European Research Council (ERC) Consolidator Grant 'vRNP-capture' 101001634 and the MRC grants MR/R021562/1, MC_UU_00034/2 and MC_UU_00034/5. J.H.

gratefully acknowledges EMBL's Infection Biology Transversal Theme Synergy Grant and support from the Deutsche Forschungsgemeinschaft (DFG) via grants HE7291/1-1 and HE7291/8-1.

## Author contributions

Conception/design: L.A., K.H., A.C., J.H; Data acquisition: L.A., K.H., V.R., L.G., S.A., I.H., B.S., P.M., F.G., A.L.; Material contribution: M.L., S.C., K.R., G.M., N.R.C.; Data analysis: L.A., K.H., L.I., L.G., K.L., F.G., A.L., A.C., J.H.; Data interpretation: L.A., K.H., L.I., A.L., F.A., A.C., J.H.; Writing, original draft: L.A., K.H., A.C., J.H.; Writing, editing, all authors; Funding acquisition: A.C., J.H.: Supervision, S.C., A.L., M.H., F.A., A.C., J.H.

## Funding

## Competing interests

The authors declare no competing interests.
