## [Peer Review File · Nature Communications]

The molecular dissection of TRIM25's RNA-binding mechanism provides key insights into its antiviral activityREVIEWER COMMENTS

Reviewer #1 (Remarks to the Author):

In this manuscript, Alvarez and colleagues perform a systematic characterization of the TRIM25 biophysical and functional properties, with a specific focus on the identification of the molecular determinants of its RNA-binding activity and its functional consequences. Using a combination of quite advanced biophysical methods, including Isothermal Calorimetry, NMR and CLIR MS/MS, they unequivocally identify a motif within the CC domain of TRIM25 as necessary and sufficient to mediate RNA binding, without affecting the overall folding or stability of the protein. Furthermore, performing iterative ITC and NMR experiments, as well as SEC-SAXS experiments on mutants conservatively mutated on 4 specific residues (CC-PRY/SPRY) in the presence of known targets of TRIM25 (pre-let-7) they identify an unexpected cooperative binding mechanism mediating RNA binding. Finally, combining the information gathered to engineer a fully RNA-binding deficient TRIM25 mutant (TRIM25-m9), they identify the specific structural (stem loops) requirements and sequence preference mediating TRIM25 binding to RNA, and show that this interaction is functionally required to restrict replication of a prototypic RNA(+) virus. Altogether, these experiments establish a new paradigm for TRIM25 RNA-binding activity, including evidence supporting a dual binding mode, regulating preferential affinity towards ssRNA vs. dsRNA.

Overall, the study is a muscular tour-de-force both from a technical and biophysical standpoint, and provides a remarkable example of integrative analysis combining biophysical *in vitro* and *ex vivo* experiments with functional data *in cellulo*. Data are clearly presented and results provided in light of the massive amount of literature existing for TRIM25 across domains (nucleic acid binding, innate immunity and restriction of viral replication). Furthermore, the last experiments supporting a specific recruitment of TRIM25 to viral replication organelles via its RNA-binding domain open multiple interesting questions on the mechanisms regulating its activity beyond ubiquitination. I have only a few comments related to the extent of validation/biological follow-up of some of these observations with respect to restriction of viral replication (see below).

Minor points:

- As clarified above, the study already presents substantial orthogonal validation efforts with functional data *in cellulo*, however it would be interesting to challenge the TRIM25-m9 mutant with respect to its binding to ZAP to ultimately address potential mechanisms underlying its recruitment to vROs. Such experiments could easily be performed with the tools described in this study ((i.e. co-IP+WB or co-IP + MS upon stimulation or viral infection), allowing to close fully the circle on some of the key TRIM25 partners.
- Similarly, experiments addressing the potential interdependence between the E3 ligase activity and RNA-binding might ultimately shed light on the functional cascade leading to TRIM25 activation. One of the unexpected findings of this study is the observation that E3 ligase activity and RNA binding activity are equally important for TRIM25 ability to restrict virus growth. Is the m9 mutant E3 ligase activity completely unaffected when compared to WT or the E3 dead mutant?
- Line 104. Fig 1, panel c is missing.

Reviewer #2 (Remarks to the Author):

The manuscript by Alvarez et al describes a structural and biophysical characterization of the RNA binding properties of TRIM25 followed by cell-based analysis of RNA binding and anti-viral functions. Overall, the manuscript is well presented and scientifically sound, although some sections could be extended to describe the results in more details.

Major Comments:

1: lanes 121-130 and Extended data Figure 1:

The results presented in Extended Figure 1 should be better interpreted and described. For example, The NOESY of Lnczc3 in Supp Fig. 1C doesn't show NOES suggesting that it is not folded into a SL, yet binding site 2 of TRIM25 is affected by the addition of this RNA, which doesn't seem to be compatible with the conclusions that binding site 2 interacts with stem-loop regions. What about the 28-mer RNA? Does it form a SL (there are no NOESY data for this RNA)? The CSPs are stronger for pre-Let7 and 28-mer than for the other two RNAs, suggesting that pre-Let-7 and the 28-mer are better binders. Are there any structural or sequence similarities between these 2 RNAs? It would be interesting to show a NOESY of the 28-mer RNA as well as ITC data of TRIM25 with the 28-mer RNA.

It is surprising that ITC shows that the binding of TRIM25 to DENV is stronger than to pre-Let7, while the CSPs are much weaker. How do the authors explain this?

2: Overall, the stoichiometries obtained by ITC are strange. With PRY, N varies between 0.7 and 0.25 and with CC it is 0.5 or n.d. Is there any reason for that? Would that be a concentration problem, or do the authors think that these are not 1:1 complexes? This should be discussed.

3: lane 163, Extended data Fig. 2C. Could the authors show an overlay of the SEC traces in addition to the table?

3: Lanes 191-192, Extended data Fig. 3a:

The ITC curves of the CC_PRY construct with Lnc3h7a and DENV are very atypical and very different from the curve observed with pre-Let7a. Can the authors comment on the difference in behaviour of the complexes between TRIM25 and different RNAs?

4: Lane 205, 206, 218:

From NMR and SAXS data, the authors conclude that the RNA enhances the intramolecular interaction between the CC and the PRY domain. Another possible explanation would be that the RNA is sandwiched between the CC and the PRY domain inducing a rigidification of the interdomain motions, while the two domains don't interact with each others. Could the authors comment on this possibility?

5: Lanes 222-224:

The authors created a RNA-binding deficient mutant (m9) and proved the lack of RNA binding by ITC. Could the authors analyse this mutant by NMR and SEC-MALS to demonstrate that the mutant is still properly folded (at least the PRY domain) and able to homodimerize?

6- Lanes 268-280, Fig. 4

Using iCLIP2, the authors identified the motifs AGAA and UGG as enriched in crosslinking sites. However, it is not clear whether this analysis was done on the uninfected or SINV-infected datasets. Are these motifs enriched in both datasets? It would be interesting to do the analysis presented in Fig. 4 d, e and f independently for each dataset and then compare the outcome. This is important because the data presented in Fig. 4c show that TRIM25 bind different regions of the RNA (3'UTR in uninfected, and CDS in infected).

The data presented in Extended data Fig. 4d are not clear and don't seem to correlate with the statement made on Lanes 273-275. Which other RBPs were analysed?

7- Lanes 298, 299, Fig. 4h:

Could the authors include the sequences of the SINV RNA bound by TRIM25 in a supplementary Figure or table?

8- Discussion (Lanes 343, 363, 364):

To illustrate and summarize the findings of this manuscript, it would be informative to include a Supplementary figure containing the iCLIP2 motifs identified (AGAA and UGG) together with the mapping of these motifs in the RNAs used in biophysical methods and the SINV RNA regions identified by iCLIP2? Are A/G-rich motifs found in all the loops of these RNAs? Could the authors correlate the presence of these motifs with the affinities observed by ITC? How do the sequences of the SINV binding region correlate with the RNA sequences of pre-Let7, DENV, ...?

Minor comments:

9- lanes 102-104:

Please mention that the resonance assignment was done previously and provide the reference here. In the present manuscript, this is only mentioned in the material and methods section.

10- Lane 231: TRIM25-m9 (no RNA binding by full-length TRIM25).

This is misleading. The data reported demonstrate that these mutations abrogate the RNA binding ability of the CC-PRY domains, not the full length protein.

Reviewer #3 (Remarks to the Author):

In the manuscript titled "The molecular dissection of TRIM25's RNA-binding mechanism provides key insights into its antiviral activity", the authors characterized the TRIM25-RNA interaction and its role in antiviral activities using multiple approaches. The study is supported by well-designed and well-controlled experiments, and it provides a clear description on the newly generated RNA-binding deficient TRIM25 and the overall TRIM25-RNA interaction, both of which are valuable to the scientific community. The presentation is succinct and mostly clear. However, considering the existing literature and knowledge about TRIM25, whether the scientific advances described in the current manuscript are sufficient for publication in Nature Communications is not self-evident.

Given the wealth of information on TRIM25-RNA interaction described in the manuscript, it feels like a missed opportunity for the authors not to test their TRIM25-RNA interaction model on the TRIM25-SINV interaction from the virus side. It will also justify the statement "...and showed that its binding to SPECIFIC viral RNA regions is critical for its antiviral activity..." in the abstract. (Currently, it is an overstatement as the evidence mainly comes from the binding deficient TRIM25 mutant.)

For the revision, the authors should attempt to generate mutant SINVs with alterations on the stem-loop structures and/or nucleotide changes in the identified TRIM25-binding region(s) aimed at disrupting TRIM25-SINV interaction based on their model. The authors should then assay these mutant SINVs to see if the alterations have the expected effects on TRIM25 binding in addition to the viral infection course. It is possible some changes might affect the general fitness of the virus and precludes further investigation. However, any insights derived from these experiments should strengthen the described TRIM25-RNA interaction model and elevate the significance of the manuscript. Furthermore, the authors need to improve the clarity and presentation in some parts of the manuscript for the revision. Please see the comments below for details.

#Main comments/Questions:

1) Fig. 4c, comparison between WT mock and WT SINV: I doubt that's the case, but does SINV

infection alters general 3'UTR usage? This is regarding whether changes of 3'UTR usage contribute to (any extent) the decrease of 3'UTR binding of TRIM25 in the meta-gene plot.

2) Line 278-280, "Of note, this consensus is very similar to the model RNAs...": Can the authors provide an illustration clearly label the nucleotides in pre-let-7 and DENV-SL that are similar to the motifs? I tried hard to look at Extended Data Fig. 4b but could not figure them out. (It also doesn't help that there are more than one motif shown in Fig. 4f...)

3) Line 310, "Importantly, rescue with TRIM25-m3 and particularly with TRIM25-m9 increased capsid abundance to SIMILAR levels as the E3-dead mutant.": I agree that m3 is similar to E3-dead but m9 clearly is higher than both m3 and E3-dead mutants in both the left panel (either at 9 or 18hpi) and the right panel (based the average fold change). Therefore, describe all 3 mutants as "SIMILAR" seems to be inaccurate. If the authors want to emphasize all 3 mutants have weaker antiviral activities, maybe adding a statistical test to the right panel showing they are not statistically significantly different would help, but the description in the text still needs to be revised to be more accurate.

4) Line 312, "...rescued with WT TRIM25 restores the antiviral function TO THE LEVELS...": It looks like a PARTIAL rescue at 18hpi (lower loading at 9hpi makes it difficult to conclude). The authors can provide more data to demonstrate that it is a FULL rescue. Otherwise, it might be more accurate to say that the antiviral function in KO cells can be partially rescued in by expressing WT TRIM25, and this is the system being utilized to test different TRIM25 mutants.

5) Line 314-316, "...chimeric viruses expressing mScarlet...or mCherry...": Please move the citation of Extended Data Fig. 5b up from Line 321 to the end of this sentence.

6) Line 360-361, "...identified A PREFERRED SEQUENCE MOTIF...": Please specify the sequence motif as there are two motifs shown in Fig. 4f.

7) Line 362-364, "...is present in previously identified RNAs...": See 2) above, please specify the motif, add an illustration and refer to the new figure.

#Figures/legends that need clarification and/or additional information.

1) Fig. 4: the abbreviation hpi is not defined.

2) Fig. 4d: the X-axis is not defined. If the crosslink site is in the center, would it be more intuitive to have the crosslink site as 0 and label the axis as -25 to +25?

3) Fig. 4f: First, the top motifs are not defined. Are they the two separate consensus motifs derived from Fig. 4e motifs? Second, the use of different colors in the plot are not defined or described. Third, the X-axis for the plot is also not defined.

4) Fig. 4h: First, the "identified motifs" are not clearly defined. Are they the motifs in Fig. 4e or Fig. 4f? If they are the consensus motifs in Fig. 4f, which one is it? Or are they both represented? Second, it might help to add 5' and 3' annotation to the hairpin structures. This figure carries a lot of important information but I do suggest the authors to improve the presentation. If possible, maybe the authors can also list the exact nucleotide sequences in the same structure in a new Extended Data Figure?

5) Fig. 5a, right panel: First, which time point are these quantifications from? 9hpi or 18hpi? Second, would the authors consider adding a statistical test between groups?

6) Extended Data Fig. 5c: The colors representing different cell lines are not described.

REVIEWER COMMENTS

Author Rebuttal to Initial comments

We thank the reviewers for evaluating our work. The reviewers raised many important and insightful questions. We believe we have now addressed these through new experiments and updated text, substantially improving the manuscript in the process.

We modified the Main Text and Supplementary Information to include the new results and interpretations made in our revisions. Below, we provide a point-by-point response to the concerns raised by the reviewers, with our responses in blue and italic. The modified text in the revised manuscript is highlighted in grey to facilitate evaluation of our revisions.

We have also changed the manuscript to comply with the editorial requests. Now there is only one Supplementary File, which will contain the previous Extended Data Figures together with the Supplementary Tables.

Reviewer #1:

In this manuscript, Alvarez and colleagues perform a systematic characterization of the TRIM25 biophysical and functional properties, with a specific focus on the identification of the molecular determinants of its RNA-binding activity and its functional consequences. Using a combination of quite advanced biophysical methods, including Isothermal Calorimetry, NMR and CLIR MS/MS, they unequivocally identify a motif within the CC domain of TRIM25 as necessary and sufficient to mediate RNA binding, without affecting the overall folding or stability of the protein. Furthermore, performing iterative ITC and NMR experiments, as well as SEC-SAXS experiments on mutants conservatively mutated on 4 specific residues (CC-PRY/SPRY) in the presence of known targets of TRIM25 (prelet-7) they identify an unexpected cooperative binding mechanism mediating RNA binding. Finally, combining the information gathered to engineer a fully RNA-binding deficient TRIM25 mutant (TRIM25-m9), they identify the specific structural (stem loops) requirements and sequence preference mediating TRIM25 binding to RNA, and show that this interaction is functionally required to restrict replication of a prototypic RNA(+) virus. Altogether, these experiments establish a new paradigm for TRIM25 RNA-binding activity, including evidence supporting a dual binding mode, regulating preferential affinity towards ssRNA vs. dsRNA.

Overall, the study is a muscular tour-de-force both from a technical and biophysical standpoint and provides a remarkable example of integrative analysis combining biophysical in vitro and ex vivo experiments with functional data in cellulo. Data are clearly presented, and results provided in light of the massive amount of literature existing for TRIM25 across domains (nucleic acid binding, innate immunity and restriction of viral replication). Furthermore, the last experiments supporting a specific recruitment of TRIM25 to viral replication organelles via its RNA-binding domain open multiple interesting questions on the mechanisms regulating its activity beyond ubiquitination. I have only a few comments related to the extent of validation/biological follow-up of some of these observations with respect to restriction of viral replication (see below).

We thank the reviewer for this overall very positive assessment.

As clarified above, the study already presents substantial orthogonal validation efforts with functional data in cellulo, however it would be interesting to challenge the TRIM25-m9 mutant with respect to its binding to ZAP to ultimately address potential mechanisms underlying its recruitment to vROs. Such experiments could easily be performed with the tools described in this study (i.e. co-IP+WB or co-IP + MS upon stimulation or viral infection), allowing to close fully the circle on some of the key TRIM25 partners.

We appreciate the suggestion made by the reviewer to further explore the interaction between the TRIM25-m9 mutant and ZAP. We agree that this is particularly important in the context of elucidating potential mechanisms underlying its recruitment to vROs. We acknowledge the significance of this line of investigation in shedding light on the role of TRIM25 in viral infection. We would like to highlight that we are already working on the role of TRIM25 RNA binding in the interaction with ZAP as well as other interactors of TRIM25. For that reason, we believe that addressing this aspect would be better suited for a separate study where it can be thoroughly explored in depth, without deviating from the central narrative of our current manuscript.

Similarly, experiments addressing the potential interdependence between the E3 ligase activity and RNA-binding might ultimately shed light on the functional cascade leading to TRIM25 activation. One of the unexpected findings of this study is the observation that E3 ligase activity and RNA binding activity are equally important for TRIM25 ability to restrict virus growth. Is the m9 mutant E3 ligase activity completely unaffected when compared to WT or the E3 dead mutant?

We thank the reviewer for this comment, which has provided us with valuable insight into the functional properties of the TRIM25-m9 mutant. To address this point, we performed additional experiments to assess whether the RNA-

binding mutation affects E3 ligase activity compared to TRIM25-WT. The E3 ligase activity of the m9 mutant was assessed by in vivo ubiquitination assays. Detailed methods and results are shown in the Supplementary Figure 4 (panel b) and described in the Methods section (page 18, lines 520-533). Our results indicate that the m9 mutant retains E3 ligase activity when compared to the E3-dead mutant, as the auto mono- and poly-ubiquitination bands can be observed for the m9 but not for the E3-dead mutant. However, the E3 ligase activity of TRIM25-m9 appears to be compromised compared to TRIM25-WT, since the intensity of the band corresponding to mono-ubiquitination is substantially weaker relative to the non-ubiquitinated when compared to the WT protein (see Supplementary Figure 4 panel b). In addition, the intensity of the smear corresponding to the poly-ubiquitination is less intense for m9. These results confirm that RNA binding plays a role for TRIM25s E3 ligase activity and that mutations disrupting its RNA-binding activity reduces ligase activity, as previously suggested by Choudhury et al., 2017 [PMID: 29117863] and Sanchez et al., 2018 [PMID: 30342007]. We have included this in the Results section (page 9, lines 213-215).

Line 104. Fig 1, panel c is missing.
This has been amended.

Reviewer #2:

The manuscript by Alvarez et al describes a structural and biophysical characterization of the RNA binding properties of TRIM25 followed by cell-based analysis of RNA binding and anti-viral functions. Overall, the manuscript is well presented and scientifically sound, although some sections could be extended to describe the results in more details.

1) Lanes 121-130 and Extended data Figure 1:

The results presented in Extended Figure 1 should be better interpreted and described. For example, The NOESY of Lnczc3 in Supp Fig. 1C doesn't show NOES suggesting that it is not folded into a SL, yet binding site 2 of TRIM25 is affected by the addition of this RNA, which doesn't seem to be compatible with the conclusions that binding site 2 interacts with stem-loop regions.

We added a description of the ¹H/¹H-2D-NOESY experiments of the different RNAs in the result section (page 3, lines 80-82). Regarding the NOESY spectrum of Lnczc3h7a, while it is true that NOE cross-peaks are not readily visible, it is important to note that imino peaks (10-15 ppm) are indeed present. These imino peaks provide evidence for the formation of a stem structure within Lnczc3h7a, since they are only observable when they are protected from exchange with water. In other words, exchange is very fast when bases are unpaired and thus exposed to the water, and becomes slow when bases pair and minimise their interactions with the water. The absence of NOE cross-peaks can be attributed to the low concentration of the sample. We have added a statement to the text which clarifies these points (page 3, lines 76-80).

What about the 28-mer RNA? Does it form a SL (there are no NOESY data for this RNA)?

In response to this point, we have included the ¹H/¹H-2D-NOESY of the 28-mer in Supplementary Figure 1 Panel c, which confirms that the 28-mer tested in the revised manuscript has a double-stranded region as evident from the clear imino signals. We also added ¹H/¹H-2D-NOESY experiments for the newly tested RNAs in the revised version of the manuscript.

The CSPs are stronger for pre-Let7 and 28-mer than for the other two RNAs, suggesting that pre-Let-7 and the 28-mer are better binders. Are there any structural or sequence similarities between these 2 RNAs?

It is indeed correct that pre-let-7 and 28-mer are stronger binders of TRIM25 both showing similar K_D (~1 μ M) as obtained by ITC experiments (Figure 1e, Supplementary Table 1 and Supplementary Figure 1e). Structure analysis shows that both RNAs form a stem-loop (Supplementary Figure 1c). Both RNAs contain motif 1 (as identified from the iCLIP data). Additionally, the 28-mer also has the motif 2. To illustrate this observation, we have also added a scheme in Supplementary Figure 4h that shows the mapping of the motifs found in iCLIP2 with the sequences of the RNAs tested in vitro.

It would be interesting to show a NOESY of the 28-mer RNA as well as ITC data of TRIM25 with the 28-mer RNA.

We have included the NOESY of the 28-mer RNA in Supplementary Figure 1 Panel c in the revised manuscript. Furthermore, we have performed ITC experiments with the 28-mer RNA and the PRYSPRY domain to gain deeper insights into their binding interaction. As mentioned above, the NOESY spectrum reveals the presence of imino peaks which means that the 28-mer also has double-stranded RNA region. Additionally, our ITC experiments showed a strong affinity of the PRY/SPRY domain towards this RNA (K_D 0.9 μ M). These additional experiments provide valuable complementary data that enhance our understanding of the RNA-binding of the PRY/SPRY domain of TRIM25. We have now added the ITC results of the 28-mer in the Supplementary Fig. 1e and in the Supplementary Table 1).

It is surprising that ITC shows that the binding of TRIM25 to DENV is stronger than to pre-Let7, while the CSPs are much weaker. How do the authors explain this?

Thank you for raising this point. We acknowledge the apparent inconsistency between the binding affinities observed in the ITC experiments and the weaker chemical shift perturbations (CSPs) observed for the TRIM25-DENV-SL interaction compared to the TRIM25-pre-let-7 interaction. It's important to note that ITC measures the total heat exchanged during binding, which includes contributions not only from the formation of the protein-RNA complex, but also from any conformational changes or solvent rearrangements that occur during the binding process. Therefore, the binding affinity observed in ITC experiments reflects the overall thermodynamics of the binding event. On the other hand, CSPs are indicative of local structural changes induced by RNA binding and may not fully capture the overall strength or dynamics of the binding interaction. Thus, while the weaker CSPs for the TRIM25-DENV-SL interaction may appear to contradict the stronger binding affinity observed in the ITC experiments, this is likely to be a result of the differences and limitations of each experimental technique. However, both data sources complement well to provide different lines of evidence to interpret the results.

2) Overall, the stoichiometries obtained by ITC are strange. With PRY, N varies between 0.7 and 0.25 and with CC it is 0.5 or n.d. Is there any reason for that? Would that be a concentration problem, or do the authors think that these are not 1:1 complexes? This should be discussed.

Thank you for the comment. We are aware of the different N values obtained. Regarding the variation for the PRY/SPRY domain, we repeated the ITC experiments and determined the concentration of both protein and RNA using different methods. However, the N values obtained remained consistent with our initial findings. In addition, in the repetition of the pre-let-7:PRY/SPRY experiments, we have now obtained an N value of 0.3 for the pre-let-7 RNA, which gives the same overall N value for the three different RNAs tested. This suggests that the observed variation is not due to a concentration problem but may reflect the inherent stoichiometry of the complex formed between the PRY/SPRY domain and the respective RNA molecules. We have now updated the data in the Figure 1 panel e and in the Supplementary Table 1.

For the CC domain, the N values were not reported because the binding curves did not reach saturation, so the N values obtained from the fit for these cases (~1) had significant errors. It would have been necessary to decrease the concentration of the analyte in the cell to achieve saturation; but further reduction was not possible due to the resulting too low signal to noise ratio.

We believe that our results reflect the inherent complexity of the binding interactions studied. It is possible that the TRIM25-RNA complexes may not strictly adhere to a 1:1 stoichiometry due to factors such as multivalent binding. In light of your feedback, we have included a discussion of this in the Results section (page 4 lines 103 to 107).

It would be very informative to get the crystal structure of the complex CC-PRY/SPRY:RNA. We have made numerous attempts and obtained crystals of the complex. These diffracted poorly, preventing structure determination.

3) Lane 163, Extended data Fig. 2C. Could the authors show an overlay of the SEC traces in addition to the table?

We have now incorporated the SEC-MALS traces alongside the table in Supplementary Figure 2c.

4) Lanes 191-192, Extended data Fig. 3a: The ITC curves of the CC_PRY construct with Lnc3h7a and DENV are very atypical and very different from the curve observed with pre-Let7a. Can the authors comment on the difference in behaviour of the complexes between TRIM25 and different RNAs?

Thank you for highlighting this observation. The biphasic behaviour observed in the ITC curves for Lnc3h7a and DENV-SL (and for newly included RNAs), in contrast to the monophasic curve observed for pre-let-7, suggests differences in the binding mechanisms between TRIM25 and different RNAs. In the case of the more atypical binding curve, this could be due to more than one binding event occurring in the ITC experiments. This could be due to various factors such as RNA conformational changes. As observed by Joint et al. (Biophysical Journal, 2019, PMID: 19450485), studying protein:DNA interactions which show a biphasic curve due to DNA annealing. We hypothesize that a similar phenomenon may occur with Lnc3h7a and DENV-SL, leading to the observed biphasic behaviour in the ITC curves. To provide a more detailed discussion of this, we have included it in the results section of the revised manuscript (page 7, lines 169 to 171).

5) Lane 205, 206, 218: From NMR and SAXS data, the authors conclude that the RNA enhances the intramolecular interaction between the CC and the PRY domain. Another possible explanation would be that the RNA is sandwiched between the CC and the PRY domain inducing a rigidification of the interdomain motions, while the two domains don't interact with each others. Could the authors comment on this possibility?

Indeed, and we thank the reviewer for pointing this out. We were biased towards stabilizing the CC-PRY/SPRY interaction of the crystal structure (PDB:6FLN) by RNA binding, considering that the RNA binding residues are close to the residues involved in the domain-domain interaction. However, it might well be that the RNA also changes the interaction or even abolishes the interaction of both domains. While a rearrangement of the domain:domain interaction is likely, it remains possible that both domains still interact in presence of RNA. We have modified the scheme presented (Figure 1) in Main Figure 3, Panel c and show now two alternative models.

Figure 1: Proposed mechanisms of RNA-induced conformational change. CC and PRY/SPRY of TRIM25 interact only transiently in the absence of RNA. Upon binding of stem-loop RNA the interaction between the two domains is stabilized. CC-PRY/SPRY dimer structure (PDB: 6FLN) shown as a surface representation with two binding sites in the PRY/SPRY domain (binding site 1 coloured in red and binding site 2 coloured in blue) and the binding site in the CC domain (coloured in purple). In a second possibility, the stem-loop RNA is sandwiched between the CC and PRY/SPRY domains, which do not interact with each other.

6) Lanes 222-224:

The authors created a RNA-binding deficient mutant (m9) and proved the lack of RNA binding by ITC. Could the authors analyse this mutant by NMR and SEC-MALS to demonstrate that the mutant is still properly folded (at least the PRY domain) and able to homodimerize?

In response to the reviewer's comment, we have now included SEC-MALS traces alongside the table in Supplementary Figure 3e, showing that the CC-PRY/SPRY m9 mutant retains its dimerization ability. In addition, we have performed ^1H , ^{15}N -HSQC experiments on the m9 mutant and compared the spectra with those of the wild type (included in Supplementary Figure 3f). The comparison revealed overlapping peaks as well as some differences indicative of the mutations introduced in the m9 mutant. Nevertheless, the peak dispersion of m9 remains the same, indicative of a folded PRY/SPRY domain. Based on the SEC-MALS analysis and NMR comparison, we are confident that the m9 mutant maintains correct folding in both the CC and PRY/SPRY domains. This conclusion is supported by our studies on the individual domains, which showed that the mutations do not disrupt the folding or dimerization ability. These additional experiments provide further evidence for the structural integrity of the m9 mutant and support its suitability for studying RNA binding interactions of TRIM25. We have included a description of these results in the results section of the revised manuscript (page 7, lines 202 to 203).

7) Lanes 268-280, Fig. 4 Using iCLIP2, the authors identified the motifs AGAA and UGG as enriched in crosslinking sites. However, it is not clear whether this analysis was done on the uninfected or SINV-infected datasets. Are these motifs enriched in both datasets? It would be interesting to do the analysis presented in Fig. 4 d, e and f independently for each dataset and then compare the outcome. This is important because the data presented in Fig. 4c show that TRIM25 bind different regions of the RNA (3'UTR in uninfected, and CDS in infected).

In response to the reviewer's comment, we have now clarified both in the text and in Figure 4 and Supplementary Figure 4 which datasets we are discussing when analysing the iCLIP2 data (cellular vs viral, mock vs infected). In particular, the analysis of the percentage of paired and unpaired sequences across the binding site shown in Figure 4d was performed for the TRIM25-WT mock sample. We have now included the same analysis for the TRIM25-WT infected sample in Supplementary Figure 4f, where the same result was observed. This suggests that TRIM25 recognises stem-loop structures in cellular RNA in both mock and infected conditions.

Regarding Figure 4e, the motif analysis for the cellular RNA targets was performed for TRIM25-WT in both mock and SINV-infected datasets together. We have now clarified this in the text, and we changed the SINV-infected to

infected (inf.) to distinguish it from binding to SINV RNA and to avoid confusion. This clarification is now in the text (page 11, lines 262-268).

Regarding panel f, where we analyse the position of the motif, as suggested by the reviewer, we have now analysed the mock and infected dataset separately. For the SINV-infected TRIM25-WT dataset, motif 1 is present before the cross-linking site and motif 2 after, with a stronger preference for each motif. For the mock-infected TRIM25-WT dataset, the pattern is less prominent likely because there are fewer binding sites (2032 vs 1019). We have now included the two density profiles in the Figure 4 panel f.

The data presented in Extended data Fig. 4d are not clear and don't seem to correlate with the statement made on Lanes 273-275. Which other RBPs were analysed?

We apologise for the lack of clarity in the text relative to this analysis. We have now clarified both the analysis and the presentation of the Extended Data Figure 4d (now Supplementary Figure 4g). Specifically, we used the ENCODE database, which contains eCLIP data for 250 experiments with 150 RBPs. We analysed the percentage of paired and unpaired sequences across the binding sites and compared it with that obtained for TRIM25 in this manuscript.

To enhance the clarity of the Extended Data Figure 4d, we have now performed a clustering of the results for the 150 RBPs analysed, resulting in 5 different clusters. This clustering reveals various types of binding site characteristics. Clusters 1 to 4 have paired RNA around the cross-linking site, but they differ in what happens further away from the cross-linking site. Our analysis for the TRIM25-WT both mock and infected fall into cluster number 5, which shows a unique pattern in which the RNA is unpaired surrounding the crosslinking site. The presence of different clusters with varied distributions of paired and unpaired RNA indicates that the distribution observed in the TRIM25 datasets is not a bias of the CLIP-based methods but reflects a preference specific to the protein. We have added the description of the clustering process in Supplementary Figure 4g and in the methods section (page 20, lines 619-627).

8) Lanes 298, 299, Fig. 4h: Could the authors include the sequences of the SINV RNA bound by TRIM25 in a supplementary Figure or table?

In response to the reviewer's comment, we have now included Supplementary Table 5 with the SINV-RNA sequences that TRIM25 binds.

9) Discussion (Lanes 343, 363, 364): To illustrate and summarize the findings of this manuscript, it would be informative to include a Supplementary figure containing the iCLIP2 motifs identified (AGAA and UGG) together with the mapping of these motifs in the RNAs used in biophysical methods and the SINV RNA regions identified by iCLIP2? Are A/G-rich motifs found in all the loops of these RNAs? Could the authors correlate the presence of these motifs with the affinities observed by ITC? How do the sequences of the SINV binding region correlate with the RNA sequences of pre-Let7, DENV, ...?

We appreciate this suggestion. We have now included the mapping of the two cellular motifs in all SINV RNA binding sites in Supplementary Table 5. In addition, we have coloured the SINV RNA binding sites shown in Figure 4 panel h according to the presence of motif 1 or 2 for a clearer visualisation. Moreover, we have created a new panel (Supplementary Figure 4 panel h) in which the motifs found in iCLIP2 are mapped onto the RNAs used in biophysical methods. As observed, motif 1 (...AGAA...) is found in the loop of pre-let-7 and in the stem of the 28-mer, motif 2 (...UGGA...) in the loops of Lnczc3h7a and 28-mer and both motifs are found in the loop of DENV-SL. Regarding the correlation between the presence of the motifs and the observed affinities, it is difficult to give a definitive answer because the RNAs tested biophysically differ not only in the presence of the motifs but also in their length, size and stem composition.

To gain a clearer understanding, we designed a new RNA (called pre-let7-modified) with the same stem sequence and length as pre-let-7, but with a loop modified to contain both motif 1 and motif 2. First, we confirmed that this RNA has a stem-loop by ¹H/¹H-2D NOESY. Then we measured its affinity for TRIM25-CC-PRY/SPRY by ITC. For this RNA the affinity is slightly lower than in the presence of motif 1 alone (180 nM for pre-let7-modified vs 60 nM for pre-let-7). Moreover, the shape of the isotherm changes and is now biphasic as seen previously for the other RNAs tested. All these new data are shown in Supplementary Figure 4 panels h, i and j of the revised manuscript together with the control of no binding for this RNA to the CC-PRY/SPRY- m9 mutant (Supplementary Figure 4 panels k). The data corresponding to the ITC measurements have been added to Supplementary Table 1 and the description of these results are in the page 11, lines 270-277 of the revised manuscript.

It is important to note that an RNA binding protein (RBP) such as TRIM25 does not bind to a single specific sequence. Instead, the RBP has likely a consensus binding sequence, in which some nucleotides are exchangeable. It would be a daunting task to try to narrow down such a consensus sequence further, especially for a motif in which structure-specificity also plays an important role. Nucleotides at other positions could

compensate for deviations from the consensus. Our results provide evidence for the sequence preference of TRIM25 to be enriched in A/G nucleotides (as seen for motifs 1 and 2), but this does not mean that all sequences to which TRIM25 binds will have exactly motifs 1 or 2 on them. For example, within the binding sites on SINV-RNA (see Supplementary Table 5) there are some binding sites that do not have the motifs on them, but the sequences are A/G rich.

10) Lanes 102-104: Please mention that the resonance assignment was done previously and provide the reference here. In the present manuscript, this is only mentioned in the material and methods section.

We have corrected this and now we mention the previous assignment and the corresponding reference in page 3 lines 66-67 in the Result section.

11) Lane 231: TRIM25-m9 (no RNA binding by full-length TRIM25). This is misleading. The data reported demonstrate that these mutations abrogate the RNA binding ability of the CC-PRY domains, not the full length protein.

It is correct that our data specifically demonstrate the effect of mutations on the RNA binding ability of the CC-PRY/SPRY domains of TRIM25. However, we would like to emphasise that the literature broadly supports the notion that only these domains play a pivotal role in the RNA-binding function of TRIM25 (see below) and RNA binding to other domains has not been observed. Therefore, while our study focused on elucidating the effect of mutations within the CC-PRY/SPRY domains, the existing research supports the assertion that these mutations effectively abolish the RNA binding ability of full-length TRIM25. Also, our iCLIP2 data suggests that we have no RNA binding in cells for the m9 mutant. We have clarified this point in the revised manuscript to ensure clarity for readers (page 9, line 211-212).

Literature showing that only the CC-PRY/SPRY domains are involved in RNA binding:

1. Kwon, S. C. et al. The RNA-binding protein repertoire of embryonic stem cells. *Nat Struct Mol Biol* **20**, 1122–30 (2013).
2. Hentze, M. W., Castello, A., Schwarzl, T. & Preiss, T. A brave new world of RNA-binding proteins. *Nat Rev Mol Cell Biol* **19**, 327–341 (2018).
3. Castello, A. et al. Comprehensive Identification of RNA-Binding Domains in Human Cells. *Mol Cell* **63**, 696–710 (2016).
4. Choudhury, N. R. et al. RNA-binding activity of TRIM25 is mediated by its PRY/SPRY domain and is required for ubiquitination. *BMC Biol* (2017) doi:10.1186/s12915-017-0444-9.
5. Sanchez, J. G. et al. TRIM25 Binds RNA to Modulate Cellular Anti-viral Defense. *J Mol Biol* **430**, 5280–5293 (2018).

Reviewer #3:

In the manuscript titled "The molecular dissection of TRIM25's RNA-binding mechanism provides key insights into its antiviral activity", the authors characterized the TRIM25-RNA interaction and its role in antiviral activities using multiple approaches. The study is supported by well-designed and well-controlled experiments, and it provides a clear description on the newly generated RNA-binding deficient TRIM25 and the overall TRIM25-RNA interaction, both of which are valuable to the scientific community. The presentation is succinct and mostly clear. However, considering the existing literature and knowledge about TRIM25, whether the scientific advances described in the current manuscript are sufficient for publication in Nature Communications is not self-evident.

We respectfully disagree. Although there exists indeed a lot of literature about TRIM25, the role of TRIM25 in innate immunity is far from being clear and many inconsistencies remain. RNA binding has not been considered for a long time and only in recent years it has been repeatedly shown that TRIM25 is an RNA binding protein. Even then many studies were incomplete or just incorrect with regards to TRIM25's RNA binding mechanism. This led to faulty conclusions about the role of RNA binding in antiviral activity. Our study is the first with a thorough biophysical approach to decipher RNA binding at residue-resolution level combined with in cellulo validation. Because of that, we were able to demonstrate that RNA binding is essential for co-localization to viral factories. Moreover, we could decipher the RNA structure- and sequence specificity of TRIM25 by using a truly integrative approach.

Given the wealth of information on TRIM25-RNA interaction described in the manuscript, it feels like a missed opportunity for the authors not to test their TRIM25-RNA interaction model on the TRIM25-SINV interaction from the virus side. It will also justify the statement "...and showed that its binding to SPECIFIC viral RNA regions is critical for its antiviral activity..." in the abstract. (Currently, it is an overstatement as the evidence mainly comes from the binding deficient TRIM25 mutant.)

Based on the reviewer input, we have revised the abstract to more accurately reflect the current evidence presented in our study.

For the revision, the authors should attempt to generate mutant SINVs with alterations on the stem-loop structures and/or nucleotide changes in the identified TRIM25-binding region(s) aimed at disrupting TRIM25-SINV interaction based on their model. The authors should then assay these mutant SINVs to see if the alterations have the expected effects on TRIM25 binding in addition to the viral infection course. It is possible some changes might affect the general fitness of the virus and precludes further investigation. However, any insights derived from these experiments should strengthen the described TRIM25-RNA interaction model and elevate the significance of the manuscript. Furthermore, the authors need to improve the clarity and presentation in some parts of the manuscript for the revision. Please see the comments below for details.

We agree that there is much more to be learned from the virus perspective. Mutating the virus may provide insightful information about the TRIM25-SINV interaction and we thought very thoroughly to perform this experiment. However, we found the following problems:

- i) There are many sites to be mutated (as can be in Supplementary Table 5) and we are concerned about affecting the replication capacity of the virus when mutating all these sites simultaneously or in combination.*
- ii) We have data that other RBPs interact in those SINV sites that could be affected by these mutations and it won't be straightforward to understand if the effect we see relates to TRIM25 or to other RBPs.*
- iii) The cost of the experiment is very high due to the need for a full synthesis of a cDNA or alternatively extremely time-consuming if each site is mutated after another.*
- iv) There are ethical and risk assessment concerns regarding the potential gain of function with the introduced mutations as the virus may become refractory to TRIM25 suppression.*

Nevertheless, to address the point, we took an alternative biophysical approach, using ITC to validate an RNA sequence derived from viral RNA to which TRIM25 cross-linked. We used the binding site 1576-1626 in SINV-RNA, which is A/G rich, and extended it using the SINV sequence to include both motifs 1 and 2. This RNA has a similar structural arrangement to DENV-SL. Interestingly, binding of this RNA is the tightest of all RNAs that we have tested in vitro ($K_D = 3.5nM$). This result supports the finding that the RNA binding sites on SINV RNA detected by iCLIP2 are indeed TRIM25 binding sites. As for reviewer 2, we would like to point out that it would be a daunting task to try to narrow down such a consensus sequence further, especially for a motif in which structure-specificity also plays an important role. Nucleotides at other positions could compensate for deviations from the consensus. Our results provide evidence for the sequence preference of TRIM25 to be enriched in A/G nucleotides (as seen for motifs 1 and 2), but this does not mean that all sequences to which TRIM25 binds will have exactly motifs 1 or 2 on them. For example, within the binding sites on SINV-RNA (see Supplementary Table 5) there are some binding sites that do not have the motifs, but still contain A/G rich sequences.

1) Fig. 4c, comparison between WT mock and WT SINV: I doubt that's the case, but does SINV infection alters general 3'UTR usage? This is regarding whether changes of 3'UTR usage contribute to (any extent) the decrease of 3'UTR binding of TRIM25 in the meta-gene plot.

*To analyse whether SINV infection alters overall 3'UTR usage, we examined the binding sites present in each dataset and their identities. As shown in Figure 5a, the number of binding sites increased upon infection (1019 in uninfected mock vs. 2032 in infected). In the density plots shown in Figure 4c, the distribution of binding sites across 5'-UTRs, CDSs and 3'-UTRs on cellular target RNAs indicates that TRIM25 prefers 3'-UTRs followed by CDSs in the mock sample. This preference is reversed upon infection, resulting in higher binding to CDSs than to 3'-UTRs. However, the absolute number of binding sites in the 3'UTR remains very similar between the two conditions (even slightly higher in the infected sample, see below). Most of the new binding sites in the infected sample map to regions in the coding sequence, which explains the different peak intensities in the meta-gene plot. In other words, binding to 3' regions is similar in infected and uninfected conditions, but new binding sites emerge upon infection that map to coding regions, particularly around start and stop codons. As mentioned in the manuscript, a possible explanation for this observation is the increased availability of these sequences in infected cells due to low ribosome occupancy because of the eIF2 α phosphorylation and derived protein synthesis shutoff. See Carrasco et al. for more details about SINV protein synthesis shutoff (The Regulation of Translation in Alphavirus-Infected Cells. *Viruses* 10, 70 (2018)).*

Figure 2: Distribution of binding sites across the RNA regions in TRIM25 mock and infected samples.

We have added the figure as Supporting Figure 4e and a description of this analysis in the revised manuscript (Page 11, Lines 242-245).

2) Line 278-280, "Of note, this consensus is very similar to the model RNAs...": Can the authors provide an illustration clearly label the nucleotides in pre-let-7 and DENV-SL that are similar to the motifs? I tried hard to look at Extended Data Fig. 4b but could not figure them out. (It also doesn't help that there are more than one motif shown in Fig. 4f...)

In response to this point, we have now included in the Supplementary Figure 4 Panel h in which the motifs found in iCLIP2 are mapped onto the RNAs used in biophysical methods. As observed, motif 1 (...AGAA...) is found in the loop of pre-let-7 and in the stem of the 28-mer, motif 2 (...UGGA...) in the loops of Lnczc3h7a and 28-mer and both motifs are found in the loop of DENVSL.

3) Line 310, "Importantly, rescue with TRIM25-m3 and particularly with TRIM25-m9 increased capsid abundance to SIMILAR levels as the E3-dead mutant.": I agree that m3 is similar to E3-dead but m9 clearly is higher than both m3 and E3-dead mutants in both the left panel (either at 9 or 18hpi) and the right panel (based the average fold change). Therefore, describe all 3 mutants as "SIMILAR" seems to be inaccurate. If the authors want to emphasize all 3 mutants have weaker antiviral activities, maybe adding a statistical test to the right panel showing they are not statistically significantly different would help, but the description in the text still needs to be revised to be more accurate.

We agree with the reviewer that the cell line rescue with TRIM25-m9 exhibits higher levels of capsid than the cell lines rescue with TRIM25-m3 and TRIM25-E3-dead mutants at both 9hpi and 18hpi. The same trend is observed in the plate reader assays (viral fitness is higher for TRIM25-m9 than for the other two mutants). However, for both methods, the differences between the mutants are not statistically significant. We have revised the text to reflect this more accurately (Page 12, lines 324-326). In addition, we have moved the Western Blot results to the Supplementary File (Figure 5 Panel a now Supplementary Figure 5 panel c) as it cross-validates the plate reader assay but is redundant to the information already shown in the figure.

4) Line 312, "...rescued with WT TRIM25 restores the antiviral function TO THE LEVELS...": It looks like a PARTIAL rescue at 18hpi (lower loading at 9hpi makes it difficult to conclude). The authors can provide more data to demonstrate that it is a FULL rescue. Otherwise, it might be more accurate to say that the antiviral function in KO cells can be partially rescued in by expressing WT TRIM25, and this is the system being utilized to test different TRIM25 mutants.

To address the reviewer's concern, we first changed the WB at 9 hpi to one where sample loading is more uniform than the one previously reported. The WB results show that the capsid levels at both 9 and 18 hpi are significantly lower in the TRIM25-WT rescue cell line than in the KO cell line and comparable (even lower due to overexpression of TRIM25 WT) to the capsid levels observed for the parental line. In addition, the plate reader assay, shown in Supplementary Figure 5 panel b, indicates that the viral fitness decreases in the TRIM25-WT rescue line with similar efficiency to the parental line when compared to the KO line. Based on these observations, we conclude that the antiviral function of TRIM25-WT is fully restored. We have clarified this in the revised manuscript (Page 12, Line 312-313).

5) Line 314-316, "...chimeric viruses expressing mScarlet...or mCherry...": Please move the citation of Extended Data Fig. 5b up from Line 321 to the end of this sentence.

We have moved this citation (Line 312, page 12).

6) Line 360-361, "...identified A PREFERRED SEQUENCE MOTIF...": Please specify the sequence motif as there are two motifs shown in Fig. 4f.

In response to this point, we have clarified this in the text (Page 17, Line 363-365) and modified Figure 4, Panels e and f, to make it clearer.

7) Line 362-364, "...is present in previously identified RNAs...": See 2) above, please specify the motif, add an illustration and refer to the new figure.

In response to this point, we have clarified this in the text and cite the new panel (Supplementary Figure 4 Panel h) in the revised manuscript.

#Figures/legends that need clarification and/or additional information.

1) Fig. 4: the abbreviation hpi is not defined.

We corrected this in the legend to Figure 4 (results section).

2) Fig. 4d: the X-axis is not defined. If the crosslink site is in the center, would it be more intuitive to have the crosslink site as 0 and label the axis as -25 to +25?

We corrected this with Figure 4d and f (results section) and Supplementary Figure 4.

3) Fig. 4f: First, the top motifs are not defined. Are they the two separate consensus motifs derived from Fig. 4e motifs? Second, the use of different colors in the plot are not defined or described. Third, the X-axis for the plot is also not defined.

We have modified the figure and now the X-axis is defined and we have improved the legend of Figure 4f and explained the unclear points.

4) Fig. 4h: First, the "identified motifs" are not clearly defined. Are they the motifs in Fig. 4e or Fig. 4f? If they are the consensus motifs in Fig. 4f, which one is it? Or are they both represented? Second, it might help to add 5' and 3' annotation to the hairpin structures. This figure carries a lot of important information but I do suggest the authors to improve the presentation. If possible, maybe the authors can also list the exact nucleotide sequences in the same structure in a new Extended Data Figure?

We have improved the visualization of Figure 4. We have clarified panel h by adding the 5' and 3' annotation and colouring the motif accordingly to the colour code established (green for motif 1 and orange for motif 2). Also, we have provided the SINV-RNA sequences that TRIM25 binds in Supplementary Table 5. In addition, we have included a new panel in Supplementary Figure 4 in which the exact nucleotide sequences are plotted in the same structure and the motifs are shown.

5) Fig. 5a, right panel: First, which time point are these quantifications from? 9hpi or 18hpi? Second, would the authors consider adding a statistical test between groups?

We have corrected this and now it is clarified in the figure legend that the quantification corresponds to 18hpi and we have added the statistical tests.

6) Extended Data Fig. 5c: The colors representing different cell lines are not described.

We have corrected this and now the colours are described in the figure legend.

REVIEWERS' COMMENTS

Reviewer #1 (Remarks to the Author):

I commend the authors for their efforts in the revision of this manuscript. The specific concerns were addressed experimentally, and the overall manuscript is improved also with respect to the clarification of some of the methods and additional experiments to the other reviewers comments. I think this study provides important information to the community with respect to TRIM25 enigmatic mode-of-action.

Reviewer #2 (Remarks to the Author):

The authors have satisfactorily addressed and the issues that I raised

Reviewer #3 (Remarks to the Author):

The authors' efforts to address all the comments are appreciated, and the revised manuscript is substantially improved compared to the original one, especially regarding the presentation and clarity. I have a small suggestion below, but all my comments/concerns are satisfactorily addressed in the revised manuscript and I would like to recommend publication.

#Suggestion:

It's great that the authors took the biophysical approach to validate TRIM25-SINV interaction using RNA_50_motif1&2. I think it is an important experiment. As a result, I tried to figure out how exactly did the author design this oligo but I couldn't find the information. I'm not sure how exactly "...extended it using the SINV sequence to include both motifs 1 and 2..." was achieved. If the authors could provide more information to help the readers fully understand the experiment either in the method or supplemental, it will be very helpful.

REVIEWERS' COMMENTS

We thank again the reviewers for evaluating our work. We modified the Main Text to address the reviewer's request. Below, we provide a point-by-point response to the concerns raised by the reviewers, with our responses in blue and italic. The modified text in the revised manuscript is marked highlighted in grey to facilitate evaluation of our revisions.

Reviewer #1 (Remarks to the Author):

I commend the authors for their efforts in the revision of this manuscript. The specific concerns were addressed experimentally, and the overall manuscript is improved also with respect to the clarification of some of the methods and additional experiments to the other reviewers comments. I think this study provides important information to the community with respect to TRIM25 enigmatic mode-of-action.

Reviewer #2 (Remarks to the Author):

The authors have satisfactorily addressed and the issues that I raised

Reviewer #3 (Remarks to the Author):

The authors' efforts to address all the comments are appreciated, and the revised manuscript is substantially improved compared to the original one, especially regarding the presentation and clarity.

I have a small suggestion below, but all my comments/concerns are satisfactorily addressed in the revised manuscript and I would like to recommend publication.

#Suggestion:

It's great that the authors took the biophysical approach to validate TRIM25-SINV interaction using RNA_50_motif1&2. I think it is an important experiment. As a result, I tried to figure out how exactly did the author design this oligo but I couldn't find the information. I'm not sure how exactly "...extended it using the SINV sequence to include both motifs 1 and 2..." was achieved. If the authors could provide more information to help the readers fully understand the experiment either in the method or supplemental, it will be very helpful.

We appreciate the suggestion made by the reviewer and we have added a longer explanation in the revised manuscript (page 7, Lines 293 to 296).